# Proteomic analysis of hippocampus reveals metabolic reprogramming in a piglet model of mild hypoxic ischemic encephalopathy

Dawn B. Lammert[1], Regina F. Fernandez[1,2], Xiuyun Liu[3,4], Jingyao Chen[2,5], Raymond C. Koehler[3], Susanna Scafidi[3]*, Joseph Scafidi[1,2]*

1 Department of Neurology, Johns Hopkins University School of Medicine, Baltimore, Maryland, United States of America, 2 The Michael V. Johnston Center for Developmental Neuroscience, Kennedy Krieger Institute, Baltimore, Maryland, United States of America, 3 Department of Anesthesiology and Critical Care Medicine, Johns Hopkins University School of Medicine, Baltimore, Maryland, United States of America, 4 Tianjin University, Tianjin, China, 5 Department of International Health, Bloomberg School of Public Health, Johns Hopkins University, Baltimore, Maryland, United States of America

* jscafid2@jhmi.edu (JS); sscafid2@jhmi.edu (SS)

## Abstract

Neonatal hypoxic-ischemic encephalopathy (HIE) remains a leading cause of long-term neurologic morbidity. Fifty percent of HIE cases are mild and do not have clearly defined therapeutic interventions. Emergent evidence now demonstrates that up to 25% of children with mild HIE suffer motor and developmental delay by 18 months and 35% have cognitive impairments by age 5 years. Interestingly, the hippocampus, which is responsible for learning and memory, does not show overt injury but does demonstrate volume changes on imaging that correlate with cognitive and behavioral outcomes. Although there is extensive data regarding pathophysiological changes following moderate and severe HIE, there is a paucity of understanding regarding the extent, duration, and compensatory adaptations in the mild neonatal HIE brain. We performed hippocampal proteomic analysis using a swine model of mild neonatal hypoxia-asphyxia. Hippocampi were collected at 24 or 72 hours after injury, and proteomics was performed by liquid chromatography tandem mass spectrometry (LC-MS/MS). Pathway analysis demonstrated that several metabolic pathways are temporally regulated after mild HIE. Specifically, amino acid, carbohydrate, and one-carbon metabolism increased at 24 hours while fat metabolism and oxidative phosphorylation decreased at 24 hours. Downregulation of oxidative phosphorylation was more pronounced at 72 hours. Our data demonstrate that metabolic reprogramming occurs after mild HIE, and these changes persist up to 72 hours after injury. These results provide new evidence that mild HIE disrupts brain metabolism, emphasizing the need for a better understanding of the underlying pathophysiology of mild HIE and development of targeted therapeutic interventions for this population.

## Introduction

Neonatal encephalopathy affects approximately 2–6 per 1,000 term births and can be caused by sepsis, stroke, inborn errors of metabolism, epilepsies, and a number of other causes [1].

**Data availability statement:** All relevant data are within the paper and its Supporting Information files.

**Funding:** This study was funded by a grant from the National Institute of Neurological Disorders and Stroke (R01NS099461) JS, National Institute of Neurological Disorders and Stroke (R01NS125653) JS and SS, National Institute of Neurological Disorders and Stroke (R01NS110808) SS, National Institute of Neurological Disorders and Stroke (R01NS111230) SS, National Heart, Lung, and Blood Institute (R01HL139543) RCK, National Institute of Neurological Disorders and Stroke (R25NS065729) DBL, Johns Hopkins Hospital Physician Scientist Experiment Fund micrograant DBL, and Michael V. Johnston Family JS.

**Competing interests:** The authors have declared that no competing interests exist.

Hypoxic-ischemic encephalopathy (HIE) affects at least 1.5 per 1,000 births and is one of the leading causes of neonatal encephalopathy and long-term morbidity and mortality worldwide [1]. HIE is characterized by (1) a sentinel event (e.g., placental abruption, tight nuchal cord, shoulder dystocia, thick meconium); (2) clinical features of encephalopathy including altered level of consciousness, activity, tone, primitive reflexes, and autonomic function; and (3) abnormal blood gases demonstrating acidosis [2–4].

Research and treatment have largely focused on moderate and severe forms of HIE. The degree of neonatal encephalopathy is determined based on the modified Sarnat score [3,4]. However, fifty percent of HIE cases are categorized as mild encephalopathy and thus not eligible for therapeutic hypothermia at many intensive care centers [5]. Initially, mild HIE was assumed to have normal outcomes, but recent research has demonstrated that 25% of infants with mild HIE have abnormal outcomes by 18 months of age, including death, motor delay, and global developmental delay [6]. Thirty-five percent of infants with mild HIE have delays at 5 years of age [5]. With these newly appreciated long-term deficits, which are likely to become more evident with age and increasing cognitive and behavioral demands of adolescence and adulthood, it is now crucial to understand the pathophysiology of mild HIE.

To date, pathophysiology of moderate and severe HIE is delineated to evolve over four phases: (1) initial insult with decreased oxygen delivery, decreased ATP production, failure of active transcellular transport, excitotoxicity, and reperfusion injury; (2) a latent phase of up to 6 hours with continued injury; (3) secondary injury with near complete mitochondrial failure lasting days; and (4) tertiary injury with late cell death, gliosis, remodeling, and repair that can continue for months after the initial insult [2]. However, whether mild HIE shares similar features to moderate and severe HIE, and the extent and duration of these changes is presently unknown.

Severe HIE results in infarcts to the deep gray nuclei, including the hippocampus, thalamus, and lentiform nucleus that are evident on brain imaging [7,8]. In contrast, when mild HIE injury is present, magnetic resonance imaging (MRI) demonstrates that cortical and subcortical white matter regions (centrum semiovale, internal capsule, splenium of the corpus collosum) are preferentially involved [7]. Interestingly, the hippocampus, a brain region responsible for memory and learning, does not show overt diffusion restriction on acute MRI, but shows decreased volume compared to age and sex-matched controls in mild HIE within the first week after injury [9]. In another study, these changes in hippocampal and thalamic volumes were observed at 7 years of age and correlated with intelligence quotient (IQ) regardless of whether the children received therapeutic hypothermia in the neonatal period [10]. These clinical studies suggest that while the hippocampus does not demonstrate overwhelming cell death in the acute post-injury period, it is vulnerable to injury in mild HIE. Due to the limited number of studies focused on mild HIE, there is a paucity of data regarding the extent of injury in the hippocampus following mild HIE.

HIE research has predominantly been conducted in rodent models [8,11,12]. However, the domestic pig (*Sus scrofa*) brain development is more similar to that of humans than other mammalian model animals, has a similar brain growth spurt to humans, and has similar sulcation and gyration [13–15]. Using a porcine model of mild HIE, we aimed to determine the proteomic changes in the hippocampus and delineate pathways which are affected temporally after injury.

## Materials and methods

### Piglet model of mild HIE

All procedures were approved by the Johns Hopkins University Animal Care and Use Committee, and all procedures complied with the United States Public Health Service Policy on

Humane Care and Use of Laboratory Animals and Guide for the Care and Use of Laboratory Animals. Neonatal Yorkshire piglets (*S. scrofa*) (3 day old, 1–2 kg, females) were randomized to sham (anesthesia only) or neonatal hypoxia-ischemia (Fig 1A). Anesthesia was induced with 5% isoflurane via nose cone. After intubation, isoflurane anesthesia was maintained at 1.5% and 70% nitrous oxide in 30% oxygen. An arterial line was placed for blood draws and monitoring of vital signs. After baseline labs were obtained, inhaled oxygen was decreased to 13% FiO2 for 45 minutes to reach an SaO2 of approximately 50%. Subsequently, the endo-tracheal tube was clamped for 6 minutes to produce asphyxia, after which piglets were then resuscitated with 50% FiO2 for 5 minutes, then allowed to recover at 25% oxygen to achieve SaO2 of 100%. Sham piglets underwent intubation, but not hypoxia and asphyxia. Once piglets recovered, anesthesia was discontinued, animals were extubated, and then piglets were closely monitored until they regained postural ambulatory control. Post-hypoxia-asphyxia labs were also measured. In total, eleven piglets were included in the study (sham = 3; 24 hour mild HIE =4; 72 hour mild HIE=4). All piglets were able to walk and feed after resuscitation and recovery. No piglets experienced cardiac arrest or demonstrated clinical seizures. 100% of sham and hypoxia-asphyxia animals survived to the experiment endpoint. At 24 hours (sham and 24 hour recovery from hypoxia-asphyxia groups) or 72 hours post-procedure, piglets were euthanized with SomnaSol intraperitoneal injection (50 mg/kg of pentobarbital and 6.4 mg/kg of phenytoin, Henry Schein Animal Health, Dublin, OH), brains were removed, hippocampi were dissected and stored for subsequent analysis at -80˚C.

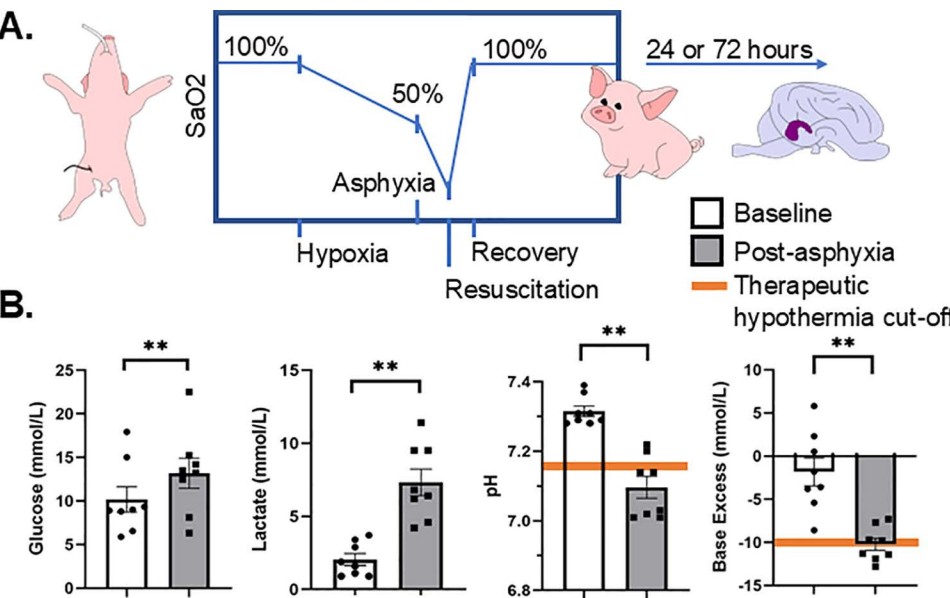

**Fig 1. Mild hypoxic-ischemic encephalopathy piglet model.** (A) Diagram of hypoxia-asphyxia protocol. Each piglet is intubated, then subjected to 45 minutes of 13% FiO2 at which point SaO2 reaches approximately 50%. The endotracheal tube is then occluded for 6 minutes to produce asphyxia. Piglets are then resuscitated with 50% FiO2 for 5 minutes, followed by 25% FiO2 until the piglet is fully recovered and ready for extubation. At either 24 or 72 hours post-procedure piglets were sacrificed. In total, eleven piglets were included in the study (sham = 3; 24 hour mild HIE = 4; 72 hour mild HIE = 4). (B) Biometric parameters of piglets who underwent asphyxia at baseline (white bar) compared to post hypoxia-asphyxia (gray bar). The orange lines denote the highest pH (7.15) and base excess (-10 mmol/L) that qualify for therapeutic hypothermia at our institution. All piglets undergoing hypoxia-asphyxia met at least one blood gas requirement.

## Proteomics analysis

**Sample collection and TCA/Acetone precipitation.** Frozen hippocampal tissue was prepared in RIPA lysis buffer with PMSF, protease inhibitor cocktail, and sodium orthovanadate (Santa Cruz, Dallas, TX) with PhosSTOP phosphatase inhibitor added (Roche, Mannheim, Germany). Tissue was homogenized using the TissueLyser II bead homogenizer (Qiagen, Hilden, Germany) followed by centrifugation and soluble aliquot saved.

Proteins (50 µg) were reduced with 50 mM dithiothreitol in 10 mM triethylammonium bicarbonate (TEAB) at 60°C for 45 minutes followed by alkylating with 100 mM iodoacetamide in 10 mM TEAB at room temperature in the dark for 15 minutes. MS interfering reagents were removed by adding 8 volumes of 10% trichloroacetic acid in cold acetone at -20°C for 2 hours to precipitate proteins. The pellet was centrifuged at 16,000 g for 10 minutes at 4°C. The TCA/Acetone supernatant was removed, and the protein pellet was washed with an equivalent 8 volumes acetone at -20°C for 10 minutes prior to centrifuging at 16,000 g for 10 minutes at 4°C. The acetone supernatant was removed from the protein pellet.

**Isobaric mass tag labeling.** The TCA/Acetone protein pellets were resuspended and digested overnight at 37°C in 100 uL 100 mM TEAB with 5 ug Trypsin/Lys-C per sample. Each sample was labeled with a unique TMTpro 16-plex reagent (Thermo Fisher, LOT # VJ313476) according to the manufacturer's instructions. All 11 TMT labeled peptide samples were combined and dried by vacuum centrifugation.

**Peptide fractionation.** The combined TMT-labeled peptides were re-constituted in 100 µL 200mM TEAB buffer and filtered through Pierce Detergent removal columns (Fisher Scientific PN 87777) to remove excess TMT label, small molecules, and lipids. Peptides in the flow through were diluted to 2 mL in 10 mM TEAB in water and loaded on a XBridge C18 Guard Column (5 µm, 2.1 x 10 mm, Waters) at 250 µL/min for 8 min prior to fractionation on a XBridge C18 Column (5 µm, 2.1 x 100 mm column, Waters) using a 0–90% acetonitrile in 10 mM TEAB gradient over 85 min at 250 µL/min on an Agilent 1200 series capillary HPLC with a micro-fraction collector. Eighty-four 250 µl fractions were collected and concatenated into 24 fractions according to Wang et al. 2011 and dried [16].

**Mass spectrometry analysis.** Peptides in each of the 24 fractions were analyzed on an Orbitrap-Fusion Lumos (Thermo Fisher Scientific) interfaced with an Easy-nLC1100 UPLC by reversed-phase chromatography using a 2%–90% acetonitrile in 0.1% formic acid gradient over 110 min at 300 nl/min on a 75 µm x 150 mm ReproSIL-Pur-120-C18-AQ column 3 µm, 120 Å (Dr.Maisch). Eluting peptides were sprayed into the mass spectrometer through a 1 µm emitter tip (New Objective) at 2.6 kV. Survey scans (MS) of precursor ions were acquired from 375–1500 m/z at 120,000 resolution at 200 m/z, with a 4e5 automatic gain control (AGC) target and maximum injection time (IT) set to auto. Precursor ions with charge states 2 to 6 were individually isolated within 0.7 m/z by data dependent monitoring and 15s dynamic exclusion, and fragmented using an HCD activation collision energy 36. Fragmentation spectra (MS/MS) were acquired using a 1.25e5 AGC, auto maximum IT at 50,000 resolution.

**Data analysis.** Fragmentation spectra were processed by Proteome Discoverer v2.5 (PD2.5, ThermoFisher Scientific) and searched with Mascot v.2.8.0 (Matrix Science, London, UK) against RefSeq2021_Sus_211119 database with 63706 entries. Search criteria included trypsin enzyme, one missed cleavage, 3 ppm precursor mass tolerance, 0.01 Da fragment mass tolerance, with TMTpro on N-terminus and carbamidomethylation on C as fixed and TMTpro on K, oxidation on M, deamidation on N or Q as variable modifications. Peptide identifications from the Mascot searches were processed within PD2.5 using Percolator at a 5% False Discovery Rate confidence threshold, based on an auto-concatenated decoy database search. Peptide spectral matches (PSMs) were filtered for Isolation Interference <30%. Relative

protein abundances of identified proteins were determined in PD2.5 from the normalized median ratio of TMT reporter ions, having average signal to noise ratios >4, from all PSMs from the same protein. ANOVA method was used to calculate the p-values of mean protein ratios for the biological replicates set up using a non-nested (or unpaired) design. Technical variation in ratios from our mass spectrometry analysis is less than 10% [17].

## Pathway analysis

Pathway analysis was first performed using g:Profiler (https://biit.cs.ut.ee/gprofiler/gost). For ease of reference, the corresponding gene name for each protein is used to refer to the respective protein. For proteomics analysis, abundance ratios were compared between sham and 24 or 72 hours survival. A p-value <0.1 was used, accepting a 10% false positive rate to aid in discovery. Gene names corresponding to proteins with significantly changed abundances were submitted for query. Species *Homo sapiens* was used, as *Sus scrofa* returned numerous unmatched gene names and *Homo sapiens* appropriately identified species homologues. All known genes were included in the statistical domain scope, g:SCS significance threshold with 0.05 user threshold was used. Kyoto Encyclopedia of Genes and Genomes (KEGG) pathways are reported. Reactome (https://reactome.org/) was used to further analyze pathways.

## Western blot

For western blots, samples were prepared in RIPA buffer and protein concentration measured by Pierce BCA assay (ThermoFisher Scientific, Waltham, MA). 30 μg of each sample were loaded and run on a 12–4% NuPAGE Bis-Tris gel (ThermoFisher Scientific, Waltham, MA) in MES running buffer at 125 V for 1 hour. Gels were transferred to a PVDF membrane (MilliporeSigma, Burlington, MA) in Tris-Glycine buffer at 100V for 1 hour. Membranes were blocked with 5% bovine serum albumin in TBS-T. Primary antibodies were used at 1:1,000 dilution and included anti-HSC 70 (Santa Cruz Biotechnology, Dallas, TX), anti-GAPDH (Cell Signaling, Danvers, MA), anti-PDHE1α (Abcam, Cambridge, United Kingdom), and anti-acetylated lysine (Cell Signaling, Danvers, MA). Secondary antibodies were used at 1:10,000 dilution and included IRDye 800CW goat anti-rabbit and IRDye 680RD goat anti-mouse (LI-COR Biosciences, Lincoln, NE). Membranes were imaged using LI-COR Odyssey CLx (LI-COR Biosciences, Lincoln, NE).

## Real time polymerase chain reaction

For real time quantitative PCR (RT-PCR), RNA was extracted from frozen samples using Trizol and RNeasy mini plus kit (Qiagen, Hilden, Germany) per manufacturer's protocol then converted to cDNA (Applied Biosystems, Waltham, MA). DNA oligonucleotide primers are listed in S1 Table (Integrated DNA Technologies, Coralville, IA). Sso Advanced Universal Syber Green reaction with 10 ng total template was used for RT-PCR reaction and detected on Biorad CFX Opus 96 (Bio-Rad, Hercules, CA). Cq data were extracted, and fold changes were calculated using the double distance method.

## Amino acid quantification

Amino acid content from frozen hippocampal tissue was determined by the Biochemical Genetics Laboratory at the Kennedy Krieger Institute. Samples were homogenized in 1X PBS, deproteinized with 10% sample volume of 35% sulfosalicylic acid dehydrate, and centrifuged at 13,000g for 10 minutes. The supernatant was collected for amino acid analysis by ion-exchange liquid chromatography using a Biochrom 30+ amino acid analyzer. Amino acid levels were reported as nmol/g of tissue.

### Blood and brain acylcarnitine quantification

Blood spots were collected from each piglet at baseline, after asphyxia, and at time of sacrifice. Acylcarnitines were extracted and analyzed as previously described [18]. A 1/8-inch diameter blood spot or approximately 50 µg of frozen hippocampal tissue per sample were used. Briefly, 100 µL internal standard was added to each sample (NSK B, Cambridge Isotopes) and subsequently extracted using methanol. Samples were butylated before being reconstituted into mobile phase acetonitrile/water/formic acid. Samples were filtered and transferred to an injection vial. Acylcarnitines were analyzed using tandem mass spectrometer (AB SCIEX QTRAP 4500, Foster City, CA) in positive ion mode, using a precursor ion scan for m/z 85, which is a product ion of butyl ester of acylcarnitines. Chemoview (AB SCIEX) was used for quantification. Blood levels of acylcarnitines are reported as nmol/mL and hippocampal levels as nmol/g.

### Statistical analysis

Data were analyzed using GraphPad Prism software (Boston, MA). Data is presented as the mean ± the standard error of the mean (SEM). Student's t-test was used to compare between two groups, and significance defined as $p<0.05$. Ordinary 1-way ANOVA was performed for multiple comparisons, and significance defined as $p<0.05$. For proteomic analysis, abundance ratios were compared between sham and 24- or 72-hours survival, and significance defined as $p<0.05$. A p-value $<0.1$ was used, accepting a 10% false positive rate to aid in discovery. Volcano plots and acylcarnitine heat maps were generated in GraphPad Prism. Proteomics heat map was generated using Heatmapper (heatmapper.ca) using average linkage clustering method and Euclidean distance measurement. ClustVis (https://biit.cs.ut.ee/clustvis/) was used for generating principle components analysis diagram [19].

## Results

### Hypoxia-asphyxia piglets demonstrate blood biochemical changes similar to human neonates with mild HIE

Clinically, mild HIE is characterized by a sentinel insult and blood gases that support a metabolic perturbation, but with mild or minimal clinical features of encephalopathy. Blood samples were collected and analyzed for degree of metabolic acidosis. At baseline, all animals, regardless of study group, demonstrated similar blood measurements, including glucose, lactate, pH, and negative base excess. Similar to newborns with mild HIE, blood gases from hypoxia-asphyxia piglets demonstrated hyperglycemia, lactate rise, metabolic acidosis, and developed a base deficit (Fig 1B). Glucose increased after hypoxia-asphyxia from 10 ± 1.5 mmol/L to 13 ± 1.7 mmol/L ($p = 0.01$). Lactate increased from 2.1 ± 0.4 mmol/L to 7.3 ± 0.9 mmol/L ($p = 0.0006$). The pH post-asphyxia decreased from 7.32 ± 0.015 to 7.10 ± 0.031 ($p = 0.0008$). The base excess post-asphyxia also decreased from -1.8 ± 1.6 to -10 ± 0.7 mmol/L ($p = 0.0003$).

### Proteomics reveals altered profiles in hypoxia-asphyxia piglets

To delineate the effect of mild HIE on hippocampus in a time-dependent manner, we performed an untargeted quantitative tagged tandem mass-spectrometry proteomics analysis at 24 and 72 hours post-injury. Out of 10,086 identified proteins, using a 5% FDR we generated a list of differentially abundant proteins. Subsequent analysis using a p-value of <0.05 and p < 0.1 allowed us to further explore potential differences among the previously identified proteins. Our analysis revealed 118 proteins (p < 0.05) with altered abundances at 24 and 72

hours post-injury compared to sham controls, while 255 proteins (p < 0.1) showed changed abundances at either 24 or 72 hours post hypoxia-asphyxia (Fig 2A). 185 proteins were altered at 24 hours, and 88 proteins were changed at 72 hours. Of these, 18 proteins showed abundance changes at both time points. The protein list generated from the analysis using p < 0.1 was utilized for pathway analysis moving forward. Principal component analysis of differentially expressed proteins demonstrated that each timepoint clustered based on a unique set of protein expression changes (Fig 2B). Heat map analysis of the 255 changed proteins demonstrated that many protein abundances are altered at 24 hours and normalize at 72 hours (Fig 2C). To better understand this trend, we analyzed proteins with altered abundances at both time points, at only 24 hours, or at only 72 hours post hypoxia-asphyxia.

## Mild HIE induces early and persistent changes in inflammatory proteins

The 18 proteins differentially expressed at both 24 and 72 hours (p<0.1) were submitted to g:Profiler (https://biit.cs.ut.ee/gprofiler/gost) for pathway analysis. No significant KEGG pathways were found. Significant Reactome pathways included "Immune System" (*FYN, DSG1, MX1, PLD4, PSMB8, KLC1*) and "Signal Transduction" (*RGS17, FYN, DSG1, SPTBN5, PSMB8, KLC1*). In addition, these 18 proteins were compared to the Velmeshev Cortex Single Cell RNA-seq data track in UCSC Genome Browser (genome.ucsc.edu) to better understand

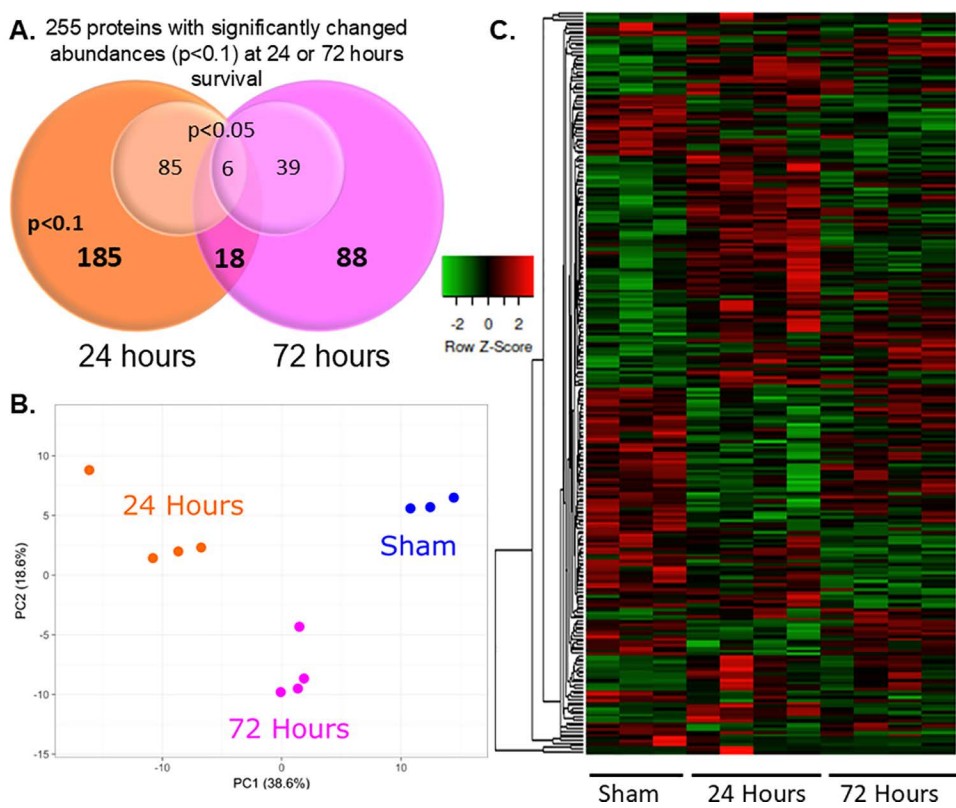

**Fig 2. Untargeted high-throughput proteomics screen.** (A) Venn diagram of differentially expressed proteins compared to sham at 24 or 72 hours post-hypoxia-asphyxia using p < 0.05 and p < 0.1. (B) Principal Component (PC) Analysis of differentially expressed proteins for each condition (p < 0.1). (C) Heat map of differentially expressed proteins at 24 or 72 hours post-hypoxia-asphyxia compared to sham (p < 0.1). Green indicates lower expression, while red indicates higher expression.

**Table 1. Up-regulated proteins at both 24 and 72 hours post-hypoxia-asphyxia.**

| Description | Cell-type expression | Gene symbol | Accession | Proteomics *Sus scrofa* | | | |
|---|---|---|---|---|---|---|---|
| | | | | Abundance Ratio: 24 Hrs/ Sham | Abundance Ratio: 72 Hrs/ Sham | Abundance Ratio P-Value: 24 Hrs/ Sham | Abundance Ratio P-Value: 72 Hrs/ Sham |
| calcium channel regulation | NA | LOC110259376 | XP_020939926.1 | 1.308 | 1.226 | 0.015 | 0.095 |
| phospholipase | **Microglia** | PLD4 | XP_020937163.1 | 1.106 | 1.126 | 0.029 | 0.055 |
| viral defense response | **Endothelium, Microglia** | MX1 | NP_999226.2 | 3.181 | 2.73 | 0.043 | 0.033 |
| axo-dendritic transport | Neuron | KLC1 | XP_005653595.2 | 1.276 | 1.214 | 0.047 | 0.098 |
| pre-mRNA processing | Neuron | PRPF6 | XP_020933272.1 | 1.839 | 2.012 | 0.056 | 0.032 |
| actin cytoskeleton | Neuron | SPTBN5 | XP_020940552.1 | 1.096 | 1.114 | 0.059 | 0.036 |
| collagen | NA | LOC102158401 | XP_020937550.1 | 1.169 | 1.173 | 0.062 | 0.044 |
| transmembrane protein | **Microglia** | TMEM119 | XP_003359182.2 | 1.397 | 1.536 | 0.079 | 0.076 |
| proteasome | **Endothelium, Microglia** | PSMB8 | NP_999100.2 | 1.348 | 1.376 | 0.086 | 0.097 |

potential cell-type specific expression [20]. Interestingly, most of the upregulated proteins at both time points are commonly involved in inflammation pathways (Table 1). Specifically, *MX1* (MX dynamin like GTPase 1) had the highest relative abundance ratio at 24 hours (3.181, p = 0.043) and 72 hours (2.73, p = 0.033). The corresponding transcript is predominantly found in endothelial cells and microglia. *MX1* is induced by interferon and involved in the cellular antiviral response [21]. *PLD4* (phospholipase D family member 4) and *TMEM119* (transmembrane protein 119) proteins were increased at both time points, and their transcripts are also predominantly found in microglia. *PLD4* is involved in phagocytosis and activation of microglia [22]. *TMEM119* is a marker of resident microglia [23]. *PSMB8* (proteasome 20S subunit beta 8) protein was also increased at both time points and is predominantly found in endothelial cells. *PSMB8* is induced by gamma interferon, is part of the immunoproteasome, and has been shown to be increased in microglia-mediated neuroinflammation [24]. These data reveal that even following mild HIE, inflammatory pathways and endothelial dysfunction are up-regulated for up to 72 hours.

In contrast, the proteins which were decreased in abundance at both time periods, are primarily associated with glia, neurons and interneurons (Table 2). *LSM11* showed the largest decrease (0.57, p = 0.02 at 24 hours and 0.57, p = 0.01 at 72 hours) and is predominantly found in parvalbumin-positive interneurons. *FYN* was decreased at both time points, and although the transcript is most highly expressed in astrocytes, as a Src family kinase, the transcript is ubiquitous in all cell types. These downregulated proteins indicate injury to hippocampal cells.

## Proteins associated with energy metabolism are temporally altered after mild HIE

To better understand acute (24 hours) and subacute (72 hours) post-injury changes in hippocampus, we compared proteomics data at either 24 hours or 72 hours post injury to sham samples. Volcano plots generated as a result of these analyses show differentially expressed proteins at each time point (Fig 3A-B).

**Table 2. Down-regulated proteins at both 24 and 72 hours post-hypoxia-asphyxia.**

| Description | Cell-type expression | Gene symbol | Accession | Proteomics *Sus scrofa* | | | |
|---|---|---|---|---|---|---|---|
| | | | | Abundance Ratio: 24 Hrs/Sham | Abundance Ratio: 72 Hrs/Sham | Abundance Ratio P-Value: 24 Hrs/Sham | Abundance Ratio P-Value: 72 Hrs/Sham |
| RNA associated | **Interneuron, Neuron** | LSM11 | XP_020932554.1 | 0.565 | 0.571 | 0.017 | 0.010 |
| Src family tyrosine kinase | **Astrocyte, Oligo-dendrocytes, Neurons** | FYN | NP_001073675.2 | 0.799 | 0.752 | 0.030 | 0.012 |
| F-box protein, ubiqui-tin proteolysis | NA | NCCRP1 | XP_020950022.1 | 0.794 | 0.741 | 0.031 | 0.011 |
| desmosome cell adhesion | NA | DSG1 | NP_001030612.1 | 0.592 | 0.475 | 0.036 | 0.014 |
| g-protein signaling | **Neuron** | RGS17 | XP_020942296.1 | 0.792 | 0.712 | 0.041 | 0.010 |
| protein phosphatase | Endothelium | PPP4R3A | XP_020955277.1 | 0.85 | 0.849 | 0.051 | 0.028 |
| unknown | NA | LOC100524118 | XP_020933178.1 | 0.809 | 0.808 | 0.056 | 0.061 |
| cyclin-dependent pro-tein kinase regulation | **Neuron** | CCNYL1 | XP_005672187.1 | 0.878 | 0.893 | 0.073 | 0.092 |
| N-acetyltransferase domain | Endothelium | NATD1 | XP_020923711.1 | 0.815 | 0.756 | 0.092 | 0.058 |

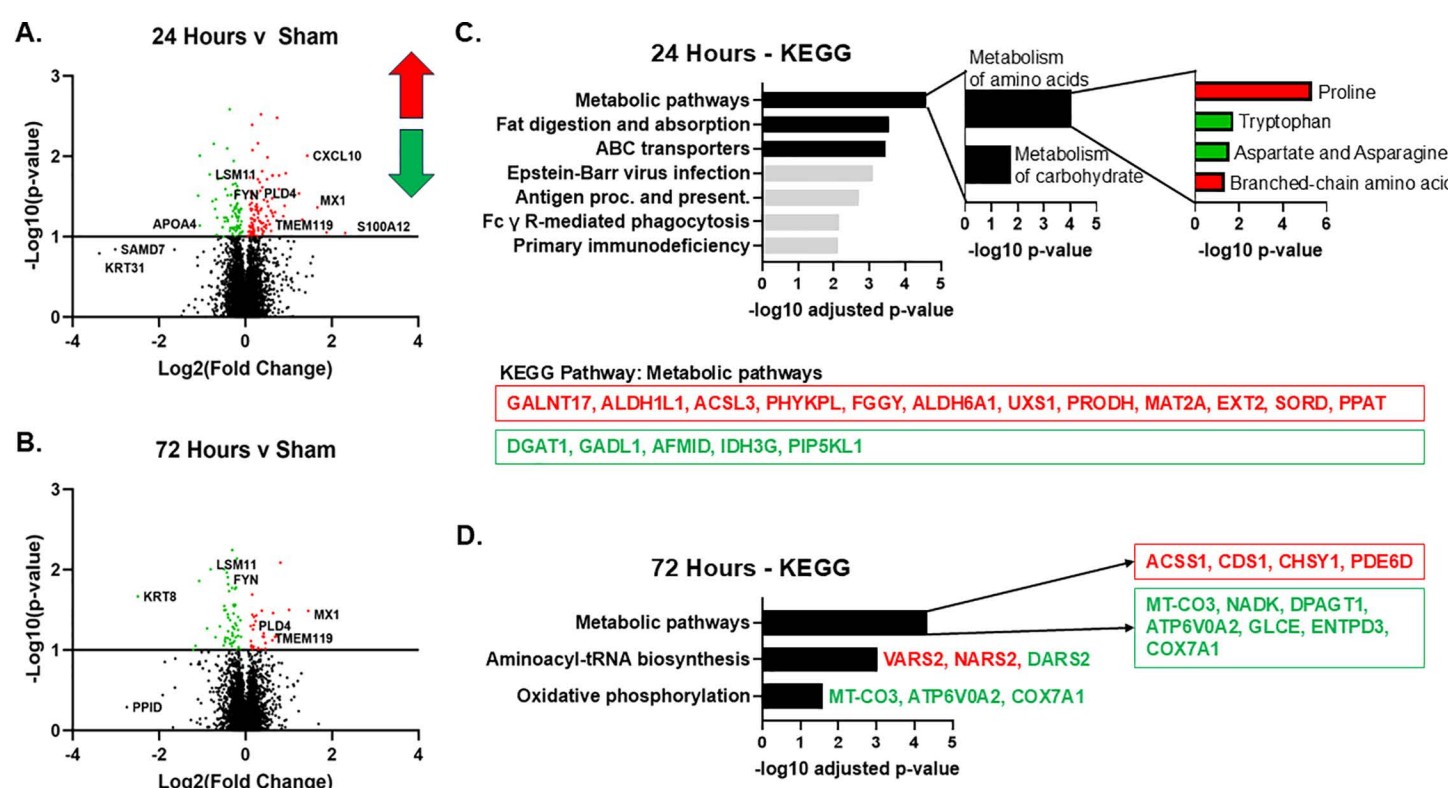

**Fig 3. Analysis of proteomics data.** Volcano plot of all proteins at (A) 24 or (B) 72 hours post-hypoxia asphyxia. Up-regulated proteins are in red, while down-regulated proteins are in green (p < 0.1). KEGG pathways for proteins altered at (C) 24-hours or (D) 72-hours post-hypoxia asphyxia are shown. Pathways of interest are emphasized in black. For the metabolic pathways, Reactome was used to better delineate sub-pathways. The top second order and third order hierarchical pathways changed at 24 hours are shown.

Our analysis showed that at 24 hours after mild HIE 185 proteins were differentially expressed compared to sham group. To better understand these changes, we further analyzed the 167 proteins with altered abundances (log2 fold change 0.14, p<0.1) unique to the 24 hour time point to determine common pathways (Fig 3C). The most significant KEGG pathways were "Metabolic Pathways," "Fat Digestion and Absorption," and "ATP-binding cassette (ABC) Transporters." Metabolic Pathways contained the proteins *GALNT17, ALDH1L1, ACSL3, PHYKPL, FGGY, ALDH6A1, UXS1, PRODH, MAT2A, EXT2, SORD*, and *PPAT* (Table 3). These mostly belong to catabolism of amino acids and branched chain amino acids, and glycosylation modifications. In contrast, proteins with decreased abundances within the Metabolic Pathways KEGG corresponded to *DGAT1, GADL1, AFMID, IDH3G*, and *PIP5KL1*, which belong to triglyceride biosynthesis, synthesis of α-ketoglutarate, and metabolism of tryptophan, aspartate and asparagine, and sulfur-containing amino acids (Fig 3C, Table 4). At 72 hours post mild HIE, we identified changes in 88 proteins, and 70 of them were uniquely changed at this sub-acute period. KEGG analysis showed that proteins with abundances only changed at 72 hours post hypoxia-asphyxia also involved "Metabolic Pathways" (Fig 3D). Other significant KEGG terms included "Aminoacyl-tRNA biosynthesis" and

**Table 3. Proteins increased at 24 hours post-hypoxia-asphyxia associated with "Metabolic Pathways" KEGG.**

| Description | Subcellular localization | Gene symbol | Accession | Proteomics *Sus scrofa* | |
| --- | --- | --- | --- | --- | --- |
| | | | | Abundance Ratio: 24 Hrs/ Sham | Abundance Ratio P-Value: 24 Hrs/ Sham |
| N-acetylgalactosaminyltransferase, membrane trafficking | golgi | GALNT17 | XP_020942008.1 | 1.223 | 0.007 |
| THF synthesis | cytosol | ALDH1L1 | XP_020953871.1 | 1.140 | 0.083 |
| lipid synthesis and fatty acid degradation | ER, golgi | ACSL3 | XP_020930228.1 | 1.093 | 0.045 |
| breaks down lysine derivative | mitochondria | PHYKPL | XP_005655024.1 | 1.666 | 0.003 |
| phosphorylates carbohydrates e.g. ribulose | cytoplasm | FGGY | XP_020953452.1 | 1.224 | 0.019 |
| valine and pyrimidine catabolism, malonate and methylmalonate semialdehyde decarboxylation | mitochondria | ALDH6A1 | XP_005656456.1 | 1.068 | 0.074 |
| synthesizes UDP-xylose, proteoglycans | golgi | UXS1 | NP_001231702.1 | 1.051 | 0.087 |
| proline degradation | mitochondria | PRODH | XP_020928191.1 | 1.298 | 0.080 |
| methionine metabolism | cytosol | MAT2A | NP_001161122.1 | 1.076 | 0.040 |
| glycosyltransferase, heparin sufate synthesis | ER, golgi | EXT2 | XP_020938817.1 | 1.121 | 0.073 |
| sorbitol dehydrogenase | exosome | SORD | NP_001231091.1 | 1.062 | 0.098 |
| purine/pyrimidine biosynthesis | cytosol | PPAT | XP_003482444.1 | 1.091 | 0.093 |

**Table 4. Proteins decreased at 24 hours post-hypoxia-asphyxia associated with "Metabolic Pathways" KEGG.**

| Description | Subcellular localization | Gene symbol | Accession | Proteomics *Sus scrofa* | |
| --- | --- | --- | --- | --- | --- |
| | | | | Abundance Ratio: 24 Hrs/ Sham | Abundance Ratio P-Value: 24 Hrs/ Sham |
| conversion of diacylglycerol and fatty acyl CoA to triacylglycerol | ER, plasma membrane | DGAT1 | XP_020944356.1 | 0.751 | 0.031 |
| carboxylic acid metabolism | cytosol | GADL1 | XP_005669383.3 | 0.754 | 0.065 |
| arylformidase, tryptophan catabolism | cytosol | AFMID | XP_020922188.1 | 0.839 | 0.044 |
| isocitrate dehydrogenase | mitochondria | IDH3G | XP_003135545.1 | 0.893 | 0.059 |
| lipid metabolism, phosphotidyl-inosilol regulation | cytosol | PIP5KL1 | XP_020925091.1 | 0.785 | 0.078 |

"Oxidative Phosphorylation." Proteins with increased abundances at 72 hours in the Metabolic Pathway KEGG included *ACSS1, CDS1, CHSY1, PDE6D* (Table 5). Decreased protein abundances in the Metabolic Pathway KEGG included *MT-CO3, NADK, DPAGT1, ATP6V0A2, GLCE, ENTPD3, COX7A1*, the majority of which are associated with mitochondrial function (Table 6). Mitochondrial aminoacyl-tRNA synthetases were also affected (DARS2 decreased at 72 hours, VARS2 and NARS2 increased at 72 hours), which are necessary for translation of oxidative phosphorylation machinery as well as other editing functions [25,26]. These data indicate that mild HIE results in metabolic perturbations of amino acid catabolism and oxidative metabolism as late as 72 hours after a sentinel event.

## Mild HIE affects amino acid levels in the brain

Since untargeted proteomic analysis showed that at 24 hours post-mild HIE metabolism of amino acid was affected, we sought to validate these findings by quantifying amino acids. Our results show that at 24 hours after mild HIE, levels of proline, tryptophan, aspartate, lysine, asparagine, and branched chain amino acids (leucine, valine and isoleucine) were increased (Fig 4) and normalized at 72 hours to levels comparable to sham animals. In addition, we observed increases in levels of several other essential (tyrosine) and non-essential (glutamate, glutamine, serine, taurine) amino acids (Table 7). The levels of mitochondria-derived ornithine, as well as hydroxy-proline (OH-proline), and alpha-amino-n-butyrate were not affected (Table 7).

## Temporal metabolic changes primarily affect non-canonical pathways

Proteins involved in metabolic pathways, and particularly oxidative phosphorylation at 72 hours, were differentially expressed. However, non-canonical metabolic enzymes were

**Table 5. Proteins increased at 72 hours post-hypoxia-asphyxia associated with "Metabolic Pathways" KEGG.**

| Description | Subcellular localization | Gene symbol | Accession | Proteomics *Sus scrofa* | |
|---|---|---|---|---|---|
| | | | | Abundance Ratio: 72 Hrs/ Sham | Abundance Ratio P-Value: 72 Hrs/ Sham |
| acetyl CoA synthetase, TCA cycle | mitochondria | ACSS1 | XP_001927148.3 | 1.342 | 0.068 |
| synthesis of phosphotidyl glycerol, cardiolipin, phosphotidylinositol | mitochondria, ER | CDS1 | NP_001037999.1 | 1.604 | 0.069 |
| chondroitin sulfate synthesis | Golgi | CHSY1 | NP_001231371.2 | 1.140 | 0.091 |
| phosphodiesterase, cilia | cytosol | PDE6D | XP_003483803.1 | 1.092 | 0.091 |

**Table 6. Proteins decreased at 72 hours post-hypoxia-asphyxia associated with "Metabolic Pathways" KEGG.**

| Description | Subcellular localization | Gene symbol | Accession | Proteomics *Sus scrofa* | |
|---|---|---|---|---|---|
| | | | | Abundance Ratio: 72 Hrs/ Sham | Abundance Ratio P-Value: 72 Hrs/ Sham |
| cytochrome c oxidase, respiratory chain | mitochondria | MT-CO3 | NP_008640.1 | 0.769 | 0.019 |
| generates NADP | cytosol | NADK | XP_020953139.1 | 0.865 | 0.050 |
| glycoprotein biosynthesis | ER | DPAGT1 | XP_003129975.1 | 0.663 | 0.051 |
| vacuolar ATPase, proton translocation | organelle and plasma membranes | ATP6V0A2 | XP_020927888.1 | 0.816 | 0.065 |
| heparan sulphate proteoglycan biosynthesis | Golgi | GLCE | XP_001927994.1 | 0.811 | 0.067 |
| extracellular ATP level regulation | plasma membrane | ENTPD3 | XP_005669433.2 | 0.930 | 0.069 |
| cytochrome c oxidase, respiratory chain | mitochondria | COX7A1 | NP_999576.1 | 0.751 | 0.071 |

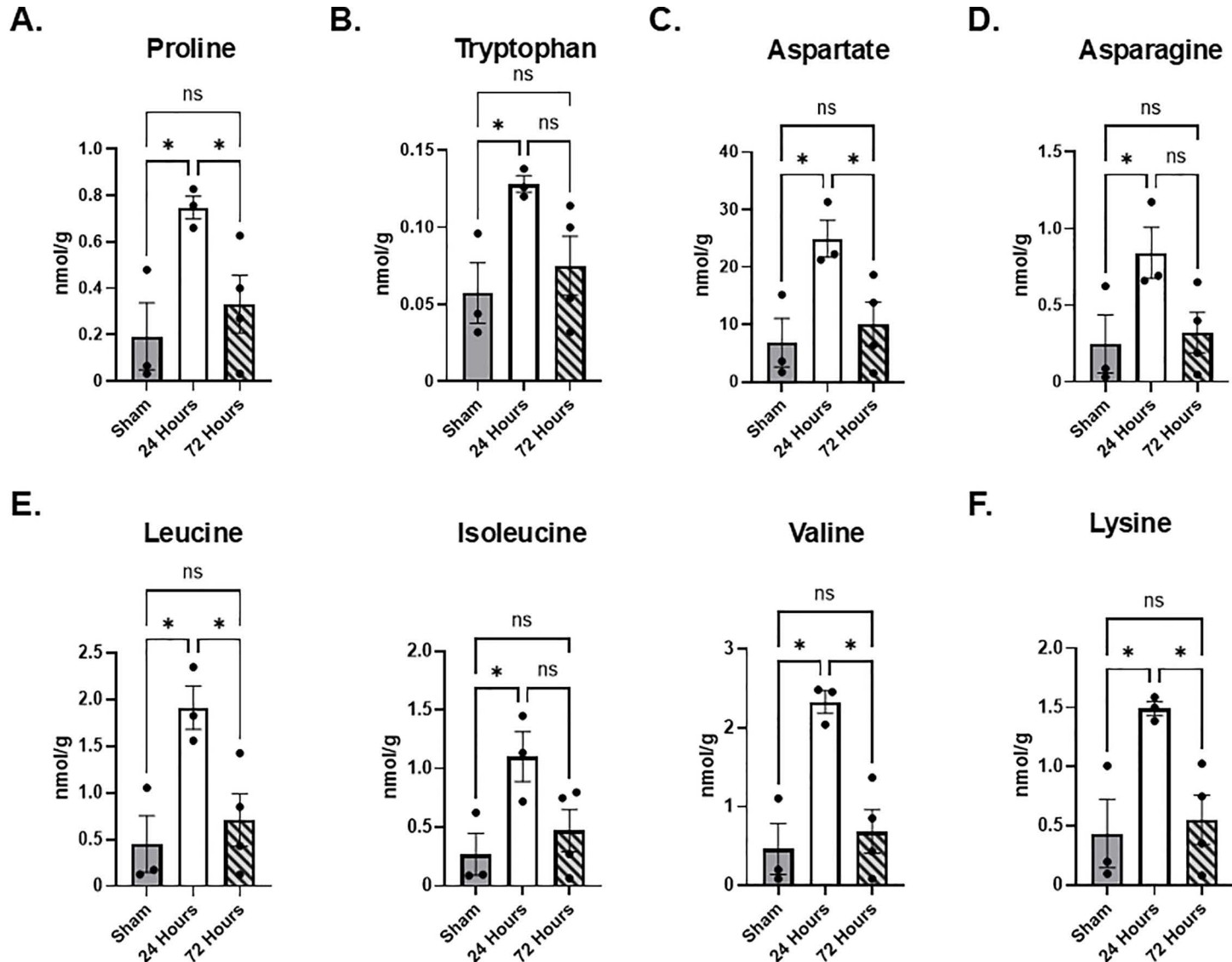

**Fig 4. Analysis of amino acid levels.** Select amino acid concentrations of interest corresponding to the proteomics metabolic pathways data are shown for sham, 24 hour post-hypoxia-asphyxia, and 72 hour post-hypoxia-asphyxia piglet hippocampus (mean, +/- SEM). (A) Proline. (B) Tryptophan. (C) Aspartate. (D) Asparagine. (E) Branched chain amino acids leucine, isoleucine, and valine. (F) Lysine. n = 3-4 per group, *p < 0.05 1-way ANOVA.

affected based on our proteomics analysis. mRNA and protein expression do not necessarily coincide (translation, posttranslational modification, etc.). Therefore, we sought to investigate mRNA expression by RT-PCR of canonical enzymes involved in glucose oxidation. Pyruvate dehydrogenase E1 subunit alpha (*PDHA1*) relative mRNA expression was significantly decreased at 24 hours compared to sham. Glyceraldehyde-3-phosphate dehydrogenase *(GAPDH)* and lactate dehydrogenase A *(LDHA)* relative mRNA expression was significantly decreased at both 24 hours and 72 hours compared to sham (Table 8).

Given the discordant change between transcription and translation of glycolysis-associated genes, we chose to verify protein expression for those proteins changed at 24 hours and 72 hours post-hypoxia-asphyxia by western blot. GAPDH and pyruvate dehydrogenase E1 subunit alpha (PDHe1α, encoded by *PDHA*) protein expression was not significantly changed

**Table 7. Amino acids in piglet hippocampus measured by ion exchange liquid chromatography.**

| Amino acid | Average (nmol/g) | | | SEM | | | 1-way ANOVA p-value | P-value | | |
|---|---|---|---|---|---|---|---|---|---|---|
| | Sham | 24Hr | 72Hr | Sham | 24Hr | 72Hr | | Sham vs 24 Hr | Sham vs 72 Hr | 24 Hr vs 72 Hr |
| Taurine | 5.46 | 21.83 | 9.63 | 3.54 | 1.76 | 3.50 | 0.0343 | 0.0219 | 0.6139 | 0.0517 |
| Phosphoethanolamine | 5.19 | 17.91 | 7.28 | 3.03 | 2.58 | 2.61 | 0.0457 | 0.031 | 0.8008 | 0.04 |
| Aspartic acid | 6.84 | 24.94 | 10.10 | 4.20 | 3.21 | 3.80 | 0.0457 | 0.031 | 0.8008 | 0.04 |
| OH proline | 0.27 | 0.70 | 0.24 | 0.18 | 0.20 | 0.08 | 0.2457 | 0.1199 | 0.8424 | 0.1432 |
| Threonine | 2.54 | 6.92 | 3.86 | 1.83 | 0.47 | 1.69 | 0.1229 | 0.0431 | 0.4711 | 0.1494 |
| Serine | 2.33 | 8.94 | 4.06 | 1.49 | 0.43 | 1.61 | 0.0343 | 0.0219 | 0.6139 | 0.0517 |
| Asparagine | 0.25 | 0.84 | 0.32 | 0.19 | 0.17 | 0.13 | 0.0343 | 0.0219 | 0.6139 | 0.0517 |
| Glutamate | 17.64 | 66.65 | 26.03 | 10.55 | 7.96 | 9.84 | 0.0343 | 0.0219 | 0.6139 | 0.0517 |
| Glutamine | 10.86 | 42.95 | 17.07 | 8.31 | 0.42 | 6.78 | 0.0343 | 0.0219 | 0.6139 | 0.0517 |
| Proline | 0.19 | 0.75 | 0.33 | 0.14 | 0.05 | 0.12 | 0.0376 | 0.0256 | 0.7043 | 0.0448 |
| Glycine | 3.03 | 12.22 | 4.10 | 1.85 | 0.11 | 1.27 | 0.0505 | 0.0431 | >0.9999 | 0.0306 |
| Alanine | 3.05 | 11.38 | 4.84 | 2.04 | 1.66 | 1.91 | 0.0848 | 0.0431 | 0.7186 | 0.0716 |
| Citrulline | 0.17 | 0.62 | 0.25 | 0.11 | 0.03 | 0.09 | 0.0376 | 0.0256 | 0.7043 | 0.0448 |
| Alpha-amino-n-butyrate | 0.07 | 0.23 | 0.11 | 0.04 | 0.05 | 0.04 | 0.1429 | 0.0679 | 0.7313 | 0.1077 |
| Valine | 0.46 | 2.33 | 0.68 | 0.32 | 0.14 | 0.28 | 0.0376 | 0.0256 | 0.7043 | 0.0448 |
| Methionine | 0.23 | 0.58 | 0.28 | 0.17 | 0.05 | 0.10 | 0.0848 | 0.0431 | 0.7186 | 0.0716 |
| Cystathionine | 0.13 | 0.64 | 0.17 | 0.08 | 0.06 | 0.05 | 0.0457 | 0.031 | 0.8008 | 0.04 |
| Isoleucine | 0.27 | 1.10 | 0.47 | 0.18 | 0.21 | 0.18 | 0.1229 | 0.0431 | 0.4711 | 0.1494 |
| Leucine | 0.45 | 1.91 | 0.71 | 0.30 | 0.23 | 0.28 | 0.0376 | 0.0256 | 0.7043 | 0.0448 |
| Tyrosine | 0.16 | 0.74 | 0.24 | 0.11 | 0.08 | 0.08 | 0.0457 | 0.031 | 0.8008 | 0.04 |
| Phenylalanine | 0.17 | 0.87 | 0.23 | 0.11 | 0.13 | 0.09 | 0.0457 | 0.031 | 0.8008 | 0.04 |
| Gaba | 5.29 | 18.18 | 8.52 | 3.77 | 2.23 | 3.35 | 0.0343 | 0.0219 | 0.6139 | 0.0517 |
| Ornithine | 0.09 | 0.26 | 0.09 | 0.05 | 0.05 | 0.03 | 0.0957 | 0.0679 | 0.9856 | 0.0488 |
| Lysine | 0.43 | 1.49 | 0.55 | 0.29 | 0.06 | 0.21 | 0.0457 | 0.031 | 0.8008 | 0.04 |
| Histidine | 0.19 | 0.76 | 0.25 | 0.14 | 0.10 | 0.11 | 0.0467 | 0.036 | 0.8993 | 0.0345 |
| Tryptophan | 0.06 | 0.13 | 0.08 | 0.02 | 0.01 | 0.02 | 0.0252 | 0.0179 | 0.527 | 0.0577 |
| Arginine | 0.16 | 0.95 | 0.28 | 0.11 | 0.22 | 0.11 | 0.0257 | 0.0179 | 0.527 | 0.0577 |

at 24 hours or 72 hours post-hypoxia-asphyxia (Fig 5A-H). Disruption of canonical pathways would be expected to produce significant injury, and proteomics and mRNA expression of canonical pathways are largely unchanged, consistent with the mild nature of injury.

### Altered proteins involved in acetyl-CoA metabolism do not lead to robust changes in lysine acetylation

A number of the proteins identified as significantly changed in the proteomics analysis can be involved in acetyl-CoA metabolism. However, mitochondrial pyruvate dehydrogenase, which generates acetyl-CoA for the TCA cycle, was unchanged by both proteomics and western blot analysis at acute (24 hours) and sub-acute (72 hours) time points after injury. Acetyl-CoA is involved in a number of processes aside from oxidative metabolism, including acetylation of proteins (e.g., histones, tubulin, mitochondrial proteins) [27]. Hence, we chose to investigate whether the changes in acetyl-CoA-associated proteins in our proteomics data lead to altered acetylated lysine. Western blot for acetylated lysine at 24 hours or 72 hours post-hypoxia-asphyxia showed no changes compared to sham controls (Fig 6).

**Table 8. Protein and mRNA expression of canonical glucose metabolism enzymes.**

| Protein (Gene name) | Pathway | Proteomics | | | | mRNA expression | | | | |
|---|---|---|---|---|---|---|---|---|---|---|
| | | 24 hour scaled abundance | 72 hour scaled abundance | 24 hour p-value | 72 hour p-value | Relative mRNA expression at 24 hr | Relative mRNA expression at 72 hr | 1-way ANOVA p-value | p-value 24 hr | p-value 72 hr |
| GLUT1 (SLC2A1) | Glycolysis | 1 | 0.976 | 0.872040679 | 0.89002083 | 1.11844268 | 1.03347367 | 0.8504 | 0.8396 | 0.9994 |
| HK1 | Glycolysis | 0.95 | 0.893 | 0.820072132 | 0.615415326 | 0.86361496 | 0.93200587 | 0.065 | 0.0409 | 0.319 |
| **GAPDH** | Glycolysis | 1.014 | 1.016 | 0.979334129 | 0.969058898 | **0.74861882** | **0.7148694** | **0.0029*** | **0.0029*** | **0.0013*** |
| **PDHA1** | Glucose oxidation | 0.993 | 1.021 | 0.999120534 | 0.675526819 | **0.8031847** | 0.86922231 | **0.0141*** | **0.0084*** | 0.0559 |
| **LDHA** | Glycolysis (favors lactate production) | 1.01 | 1.009 | 0.996470146 | 0.999569178 | **0.75235347** | **0.80521409** | **0.0246*** | **0.0171*** | **0.0494*** |
| LDHB | Glycolysis (favors pyruvate production) | 1.04 | 1.031 | 0.53955281 | 0.907831067 | 0.91486481 | 0.93433255 | | | |
| ACLY | FA metabolism | 1.028 | 1.044 | 0.994688142 | 0.994702981 | 1.10355307 | 1.21544436 | 0.5798 | 0.8729 | 0.486 |
| CPT1A | FA metabolism | 1.043 | 1.092 | 0.973945123 | 0.998518905 | 0.75146583 | 0.98928626 | 0.2022 | 0.2036 | 0.9876 |
| CPT2 | FA metabolism | 1.069 | 1.011 | 0.186785292 | 0.894893828 | 0.78410382 | 0.76411809 | 0.4822 | 0.4628 | 0.4403 |
| OXCT1 | Ketone body metabolism | 0.938 | 0.933 | 0.615515319 | 0.786912479 | 0.78452545 | 0.86500947 | 0.317 | 0.2342 | 0.4654 |
| BDH1 | Ketone body metabolism | 1.018 | 0.931 | 0.969415176 | 0.674851867 | 0.99869133 | 1.01717555 | 0.9876 | 0.9822 | 0.9979 |
| MPC1 | Glucose oxidation | 0.992 | 1.049 | 0.962423427 | 0.999999742 | 0.97625524 | 1.17255543 | 0.0573 | 0.8805 | 0.114 |
| MPC2 | Glucose oxidation | 0.979 | 1.023 | 0.996432812 | 0.997898801 | 0.96493183 | 1.1334133 | 0.5135 | 0.9539 | 0.6277 |
| PFKP | Glycolysis | 0.898 | 1.033 | 0.68421385 | 0.60783731 | 0.97533496 | 0.92507545 | 0.8865 | 0.9711 | 0.8409 |
| PFKM | Glycolysis | 1.037 | 1.011 | 0.788162865 | 0.982775764 | 0.97971072 | 1.02625903 | 0.9318 | 0.9683 | 0.9843 |
| PFKL | Glycolysis | 0.945 | 0.894 | 0.529275967 | 0.276612195 | 0.86449246 | 0.8043339 | 0.4839 | 0.5712 | 0.3915 |
| PDK2 | Glucose oxidation (inhibits) | 0.955 | 1.016 | 0.659885184 | 0.820515543 | 1.12423947 | 1.09003081 | 0.6127 | 0.5141 | 0.7085 |
| PDK3 | Glucose oxidation (inhibits) | 0.96 | 0.967 | 0.954479703 | 0.995007334 | 0.85441418 | 0.98973106 | 0.0728 | 0.0826 | 0.9779 |
| CS | TCA cycle precursors | 1.017 | 1.022 | 0.896044178 | 0.972755335 | 0.96482345 | 1.02806244 | 0.8154 | 0.9051 | 0.9598 |
| PC | Anapleurosis | 0.991 | 1.017 | 0.812652214 | 0.963935368 | 1.02642314 | 0.9976292 | 0.9497 | 0.9753 | 0.9242 |
| PPARGC1A | transcriptional regulation of metabolism | low quality protein | | | | 0.90445096 | 0.89222955 | 0.6594 | 0.6505 | 0.6025 |
| PPARGC1B | transcriptional regulation of metabolism | 1.057 | 1.031 | 0.45819025 | 0.557855189 | 0.8621792 | 1.21902561 | 0.2781 | 0.7516 | 0.5398 |
| G6PD | pentose phosphate pathway | 1.066 | 1.008 | 0.12912235 | 0.99692766 | 3.965 | 2.708 | 0.3009 | 0.2731 | 0.6607 |
| PRPS1 | nucleotide biosynthesis | 0.96 | 0.961 | 0.966788765 | 0.994805894 | 1.065 | 1.085 | 0.9458 | 0.9838 | 0.9838 |
| PRPS2 | nucleotide biosynthesis | 0.976 | 0.926 | 0.998001399 | 0.52459695 | 0.8425 | 0.85 | 0.7132 | 0.736 | 0.7557 |

## Mild HIE does not alter hippocampal acylcarnitine levels

Our proteomic data showed that mild hypoxia affected metabolic pathways including lipid metabolism-associated proteins such as DGAT1, ACSL3, APOA4, ABCA3 (Fig 3). A few studies have proposed that blood acylcarnitines may be used as biomarkers following neonatal HIE [28–30]. Acylcarnitines, generated via carnitine palmitoyltransferases, facilitate transport

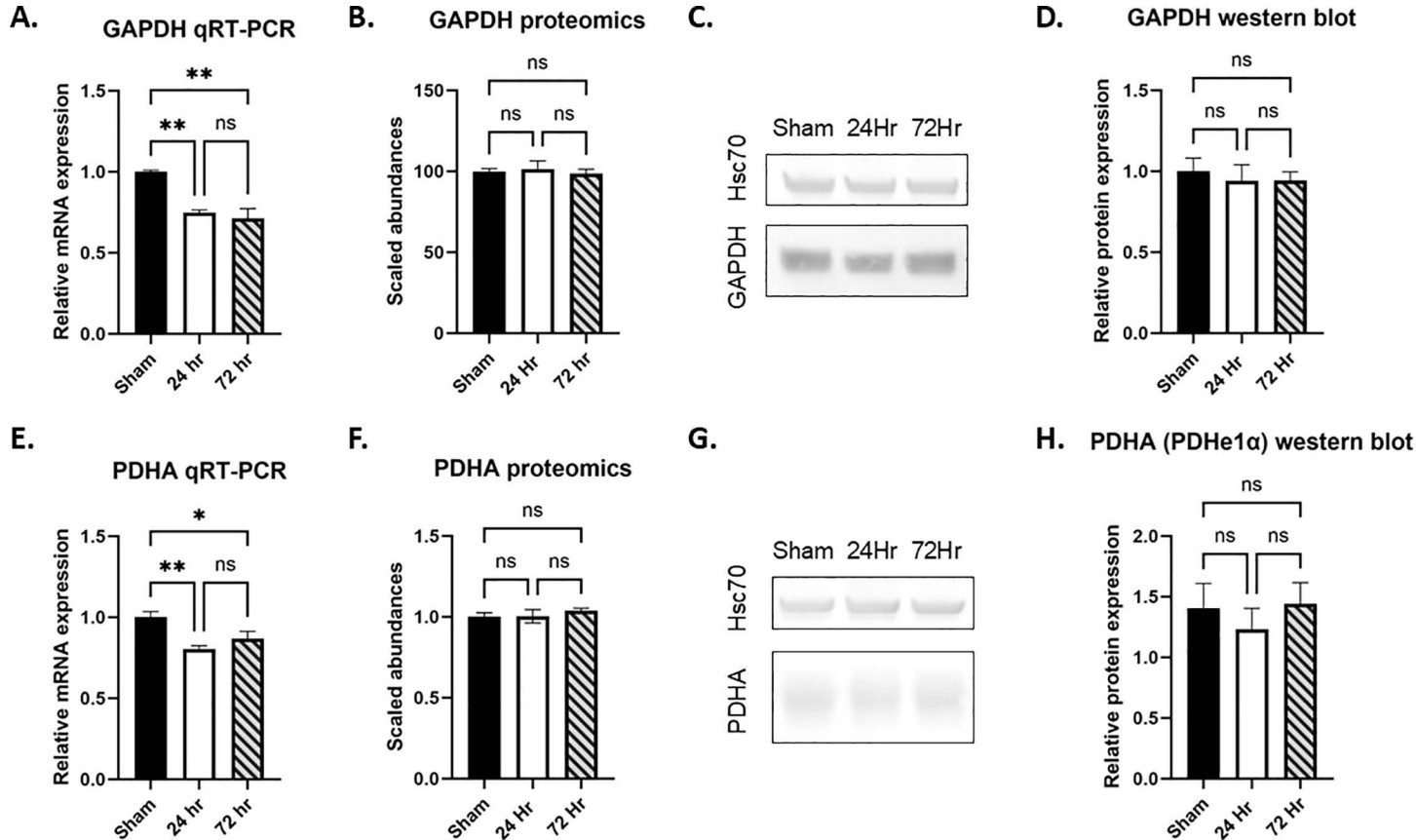

**Fig 5. Canonical enzyme expression. mRNA expression relative to sham (A, E), protein abundance by LC-MS/MS (B, F), western blot expression (C, G) and quantification of western blots (D, H) for GAPDH and PDHA are shown, respectively.** N = 3-4 per group, Ordinary 1-way ANOVA, *p <0 .05, **p < 0.01.

of fatty acids across the mitochondria membrane for subsequent β-oxidation [31]. Hence, we sought to determine whether mild HIE affects blood or hippocampal acylcarnitine profiles. Our results show that following mild HIE, blood acylcarnitines increased immediately after asphyxia, and subsequently normalized (Fig 7A, B). Total carnitine levels were not significantly different. The blood acylcarnitine level increases were primarily due to short chain acylcarnitines, specifically acetylcarnitine (C2), propionylcarnitine (C3), and butyrylcarnitine (C4) (Fig 7C). Hippocampal acylcarnitine levels were not significantly changed following mild HIE (Fig 7D, E, S1 Fig). Taken together, our results show that following mild HIE, blood acylcarnitine levels rise immediately after asphyxia, then subsequently normalize by 24 and 72 hours.

## Discussion

Currently employed HIE models successfully reproduce clinically observed moderate-severe HIE injury, which results in cardiac arrest and clinical seizures, and some piglets do not survive [13,32,33]. A mild model of HIE includes a sentinel insult and blood gases that support a metabolic perturbation, but with mild or minimal clinical features of encephalopathy. For neonates in most intensive care units, HIE severity is defined based on the eligibility criteria of early therapeutic hypothermia trials. Infants with a sentinel event must meet physiologic criteria (pH < 7.15, base deficit >10) and have a moderate to severe neurologic exam on modified

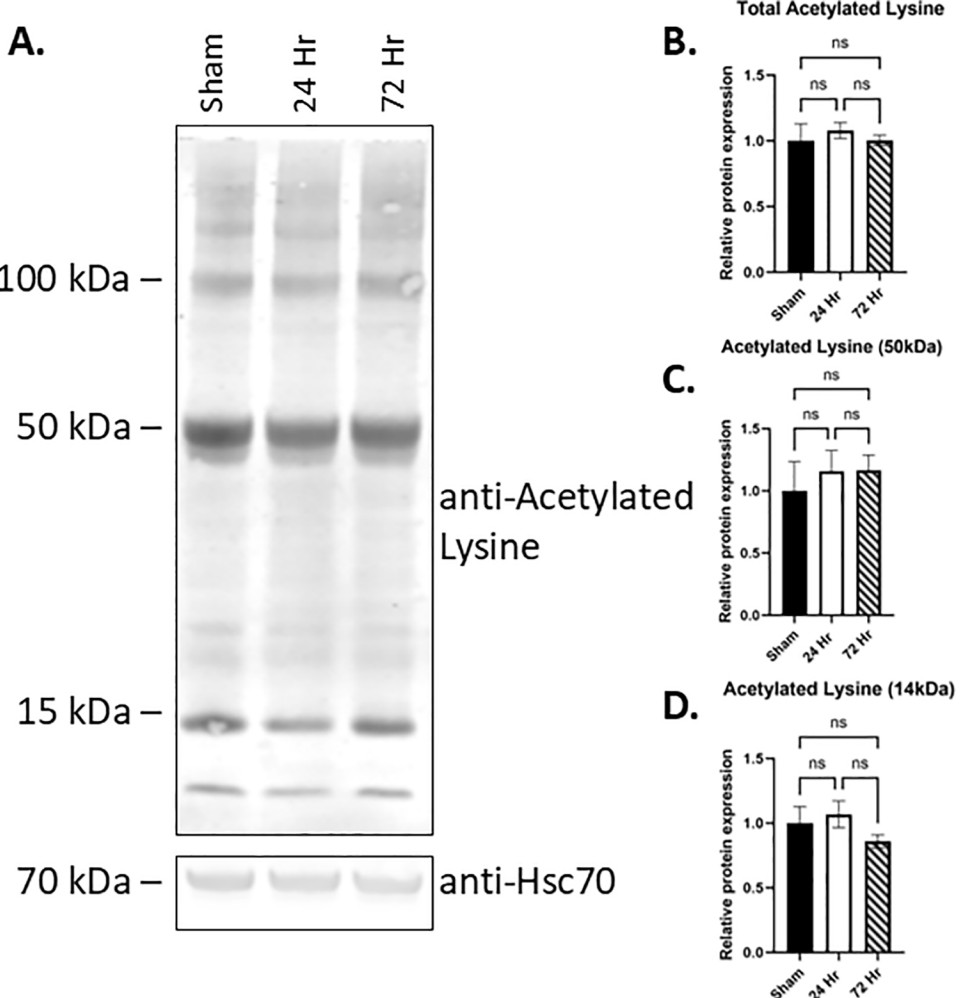

**Fig 6. Lysine acetylation after hypoxia-asphyxia does not change.** (A) Western blot for acetylated lysine of hippocampal lysates from sham, 24 hours and 72 hours post-hypoxia asphyxia. (B) Densitometry of relative total acetylated lysine normalized to HSC70. (C) Quantification of the dominant 50 kDa band. (D) Quantification of the 14 kDa band. N = 3-4 per group, Ordinary 1-way ANOVA, *p < 0.05, **p < 0.01.

Sarnat scoring [4,34]. In our model, all piglets had a sentinel event of hypoxia-asphyxia and met at least one physiologic criteria (Fig 1B), with pH range 7.01–7.22 and base deficit range 7.3–12.8 mmol/L. Clinically, piglets were able to ambulate and feed after extubation, and no piglet exhibited signs of encephalopathy or seizures. This model was modified from a previous model in which the hypoxic stress preceding the asphyxia was more severe (10% inspired O2 and 35% arterial O2 saturations instead of the currently used 13% inspired O2 and 50% arterial O2 saturations) and resulted in encephalopathy and seizures [35,36]. Therefore, this model is consistent with a mild hypoxic-ischemic injury.

Through an untargeted proteomics analysis of piglet hippocampi at 24 or 72 hours post-hypoxia-asphyxia, we show that there is evidence of hippocampal injury that persists even at 72 hours. Notably, these changes were characterized by inflammation and microglia involvement. Rodent models of moderate to severe HIE, predominantly using variations of the Rice-Vannucci model (carotid artery ligation paired with systemic hypoxia), have shown that activated microglia is one of the earliest immune responses to HIE in a region specific

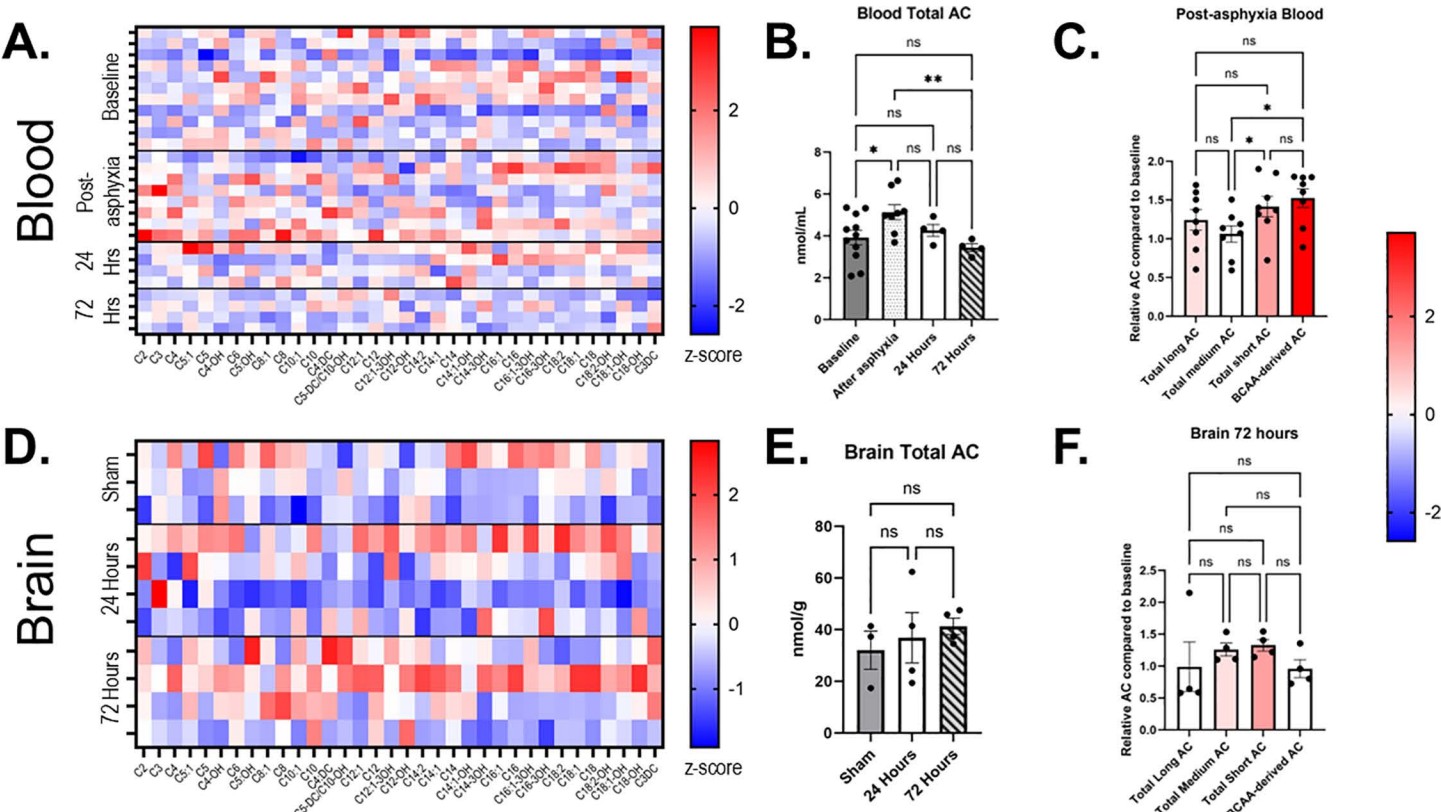

**Fig 7. Blood and brain acylcarnitines show different profiles after mild HIE.** (A) Blood acylcarnitine z-scores were calculated and presented as a heat map for each acylcarnitine species at baseline (n = 11), immediately before asphyxia (n = 8), and at sacrifice at 24 hours (n = 4) or 72 hours (n = 4). (B) Blood total acylcarnitines (AC) were calculated for each time point. (C) Post-asphyxia acylcarnitines relative to paired baseline acylcarnitines are shown and colored to parallel z-score color trends. (D) Brain acylcarnitine z-scores presented as a heat map for sham (n = 3), 24 hours post-hypoxia-asphyxia (n = 4), and 72 hours post-hypoxia-asphyxia (n = 4). (E) Brain total acylcarnitines calculated for each time point. (F) 72 hour acylcarnitines relative to sham average are shown and colored to parallel z-score trends. All calculations are ordinary one-way ANOVA, *p < 0.05, **p<0.01, and graphs show mean +/− SEM.

manner, with the earliest microglia activation occurring in the hippocampus [37]. These changes start within hours after injury and peak days after injury, with the exact timing dependent on the age of the rodent. Microglia are also important for development, trophic support, and integrity of the vascular endothelium and blood brain barrier [38]. We show that similar to moderate-severe HIE, even in mild injury, expression of proteins associated with microglia and endothelium are upregulated at 24 hours and persist to 72 hours post-hypoxia-asphyxia. Better delineation of this inflammatory response in mild HIE may lead to application of therapies that are currently being evaluated for moderate-severe HIE, such as erythropoietin [39].

Proteomic analysis has previously been undertaken for neonates with HIE from tissues other than the brain in hopes of identifying a biomarker of early HIE. Similar to blood proteomics from mild, moderate, and severe HIE human neonates, hippocampi from piglets with mild HIE showed changes in chemokine interferon inducible protein (*CXCL10*), S100 protein (*S100A12*), and apolipoproteins and their receptors (*APOA4, APOBR*) [40]. Urine proteomics analysis from neonates with HIE undertaken at 24 and 72 hours of life also identified apolipoproteins as significantly differentially expressed [41]. However, S100β, which has independently been investigated as a putative biomarker of HIE in tissues outside the brain, was not identified in our data set [42,43]. This potentially is due to tissue type investigated, ability

to detect the protein in our sample, or suggests that S100β is a marker of moderate to severe injury but not a reliable marker of mild HIE.

Our proteomic analysis of hippocampal tissue after mild HIE suggests that there is an acute and subacute energy failure. This energy failure has been previously reported only in moderate and severe encephalopathy [2]. A recent metabolomics study of cord blood in HIE and asphyxiated infants demonstrated significant changes in amino acids (asparagine, isoleucine, leucine, methionine, phenylalanine, proline, tyrosine, and valine), acylcarnitines, and glycerophospholipids [28]. Our data show changes in proteins involved in catabolism of branched-chain amino acids (leucine, isoleucine, valine), metabolism of sulfur-containing amino acids (methionine) and asparagine, and proline metabolism (*PRODH*) (Fig 3). These findings are further supported by the increases in these amino acid levels at 24 hours after injury (Fig 4). Branched-chain amino acids in particular are important sources of tricarboxylic acid cycle intermediates, and the levels were, indeed, increased at 24 hours in our hippocampal amino acid analysis (Fig 4E). Jantzie and colleagues reported that levels of GABA, alanine, and asparagine were increased in dorsal and lateral cortex of piglets at 4 hours following 2 hours of severe hypoxemia [44]. These changes were not observed in basal ganglia – structures which are commonly affected by HIE injury in both humans and laboratory animals [44]. Our results showed that amino acids levels normalized by 72 hours post-injury. In the model of severe hypoxia-ischemia, which included carotid artery ligation, amino acid analysis performed at 96 hours (4 days) post-injury did not identify any changes in the ipsilateral (site of carotid artery ligation), or contralateral cortex, basal ganglia or thalamus in the group resuscitated with 100% inspired oxygen. Recently, Shevtsova et al. reported that using an experimental model of rat hypoxia – ischemia and flow injection analysis tandem mass spectrometry (FIA-MS/MS) of dry blood spots levels of glycine, glutamine, and lysine were significantly elevated at 6 hours post injury [45]. However, how this correlates with the brain metabolic changes remains to be determined. To our knowledge, there are no clinical reports available measuring amino acids in CSF of infants following mild HIE. Interestingly, Hagberg et al. reported that CSF samples obtained at approximately 18 hours following severe asphyxia showed increased levels of glutamate and aspartate, as well as branched chain amino acids, taurine, tyrosine, and methionine, which is similar to our results at 24 hours [46].

Similar to moderate and severe encephalopathy, in the subacute period after mild HIE (72 hours post-hypoxia-asphyxia), our data show that proteins responsible for oxidative phosphorylation and mitochondrial function were affected [47]. Currently, therapeutic hypothermia undertaken over the first 72 hours following moderate-severe HIE is a standard of care and is aimed to minimize neurologic injury by decreasing metabolic dysfunction. Our results show that even mild HIE results in hippocampal injury, the area which does not demonstrate overt cell death in the neonatal period but is vulnerable to early life stress [9,10,36].

Since the proteomics showed changes in proteins responsible for mitochondrial function, we next examined the acylcarnitine profile in the blood and brain. We found that following mild HIE, total acylcarnitines in blood were significantly increased immediately after asphyxia, but subsequently normalized (Fig 7A,B). These changes were primarily driven by short chain acylcarnitines which can be derived from amino acids (Fig 7C). Brain acylcarnitines, however, did not significantly change in the hippocampus after mild HIE (Fig 7D,E, S1 Fig). Our results are similar to Dave et al., who, using a mouse model of hypoxia-ischemia, demonstrated that plasma levels of acylcarnitines peaked at 30 minutes post-injury and returned to normal by 24 hours [29]. While this group reported that in the cortex long-chain acylcarnitines (C16:0, C18:0) were increased after HIE, we observed no significant change in hippocampal acylcarnitines. Lopez-Suarez and colleagues showed that blood

levels of acylcarnitines from children with HIE showed elevated short-chain acylcarnitines, while carnitine levels were unaffected [30]. This group further correlated blood levels of C4, neuron-specific enolase levels, and MRI findings, and concluded that blood C4 level may be used as a prognostic marker in HIE. Our results showed that while blood levels of C4 increased immediately after asphyxia, this change normalized by 24–72 hours and did not correlate with hippocampal C4 acylcarnitine levels.

Although infants with mild HIE were not included in the initial studies for therapeutic hypothermia, we now know that these children have neurocognitive deficits [5,6,48]. Despite lack of evidence that therapeutic hypothermia improves outcomes after mild HIE, recognition of the adverse outcomes associated with mild HIE, combined with improved safety of therapeutic hypothermia, has led to a therapeutic drift towards treating mild HIE infants [6,49–51]. However, therapeutic hypothermia is not without risks, including potential hypotension, bradycardia, coagulopathy, fat necrosis, exposure to sedation to limit shivering, and time away from the mother. Therefore, it is imperative to determine whether therapeutic hypothermia in mild HIE targets the same metabolic and mitochondrial failure seen in moderate-severe encephalopathy, and if not, to find alternative, targeted treatments for this population.

There are notable limitations of our study. First, we were limited to a small sample size due to the exploratory nature of the study and use of a large animal model. Additionally, our proteomic analysis was limited to two time points – 24-hours (acute) and 72-hours (sub-acute). There may be other changes detectable beyond 72-hours. We have focused our studies on the hippocampus, which is not obviously injured in the first 72 hours. However, the hippocampus is vulnerable to early life insults and important for later behavioral development [9,10]. While this analysis provided new and exciting information in regard to mild HIE, it is hard to compare with currently available data from other studies of HIE, since the majority of these studies are focused on improving cell survival and decreasing cell death. Further studies are needed to determine whether sustained energy dysfunction contributes to short- and long-term pathology.

## Conclusion

In summary, this study provides new evidence that mild HIE in newborn piglet results in upregulation of inflammation and perturbations in metabolic pathways, including amino acid metabolism and mitochondrial electron transport chain in the hippocampus. Future studies aimed to better delineate pathology may provide much needed knowledge to design specific therapeutic interventions in both acute and subacute periods after mild HIE and during subsequent months and years of early child development.

## Supporting information

**S1 Table. Primer Table.** Primer sequences generated for target transcripts corresponding to the protein of interest are listed. Primer sequences were generated using PrimerBank (https://pga.mgh.harvard.edu/primerbank/) unless previously published, in which case the corresponding PubMed ID (PMID) is listed.
(XLSX)

**S1 Fig. Brain acylcarnitines.** (A-D) Heat maps of z-scores for brain acylcarnitines (AC) are shown. Individual acylcarnitine species are denoted on the x-axis. (E-H) Mean +/− SEM for each acylcarnitine group.
(TIF)

## Acknowledgments

The authors would like to the Dr. Robert Cole and Jeremy Post of the Johns Hopkins Mass Spectrometry and Proteomics Facility for their assistance with LC-MS/MS and proteomic analysis. Acylcarnitines and amino acid levels were run with the help of Dr. Lisa Kratz and the Kennedy Krieger Institute's Biochemical Genetics Lab.

## Author contributions

**Conceptualization:** Xiuyun Liu, Raymond C. Koehler, Susanna Scafidi, Joseph Scafidi.

**Formal analysis:** Dawn B. Lammert, Joseph Scafidi.

**Funding acquisition:** Dawn B. Lammert, Raymond C. Koehler, Joseph Scafidi.

**Investigation:** Dawn B. Lammert, Regina F. Fernandez, Xiuyun Liu, Jingyao Chen, Raymond C. Koehler, Susanna Scafidi, Joseph Scafidi.

**Methodology:** Xiuyun Liu, Dawn B. Lammert.

**Resources:** Raymond C. Koehler.

**Supervision:** Raymond C. Koehler, Susanna Scafidi, Joseph Scafidi.

**Writing – original draft:** Dawn B. Lammert, Regina F. Fernandez, Raymond C. Koehler, Susanna Scafidi.

**Writing – review & editing:** Dawn B. Lammert, Raymond C. Koehler, Susanna Scafidi, Joseph Scafidi.

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
