## [Decision Letter · Decision Letter 0]

7 Aug 2024

PONE-D-24-22507Proteomic analysis of hippocampus reveals metabolic reprogramming in a piglet model of mild hypoxic ischemic encephalopathyPLOS ONE

Dear Dr. Scafidi,

Thank you for submitting your manuscript to PLOS ONE. After careful consideration, we feel that it has merit but does not fully meet PLOS ONE’s publication criteria as it currently stands. Therefore, we invite you to submit a revised version of the manuscript that addresses the points raised during the review process.

We look forward to receiving your revised manuscript.

Kind regards,

Samiullah Khan, Ph. D

Academic Editor

PLOS ONE

Journal Requirements:

3. Thank you for stating the following financial disclosure: "Funding was provided to by the Johns Hopkins Hospital Physician Scientist Experiment Fund microgrant (DBL), NINDS R25 (DBL), NIH R01HL139543 (RCK), R01NS099461 (JS), R01NS125653 (JS and SS), R01NS110808 (SS), and R01NS111230 (SS)".

4. Thank you for stating the following in the Acknowledgments Section of your manuscript: "The authors would like to the Dr. Robert Cole and Jeremy Post of the Johns Hopkins Mass Spectrometry and Proteomics Facility for their assistance with LC-MS/MS and proteomics analysis. Acylcarnitines were run with the help of the Kennedy Krieger Institute’s Biochemical Genetics Lab. Funding was provided to by the Johns Hopkins Hospital Physician Scientist Experiment Fund microgrant (DBL), NINDS R25 (DBL), NIH R01HL139543 (RCK), R01NS099461 (JS), R01NS125653 (JS and SS), R01NS110808 (SS), and R01NS111230 (SS)."

Please remove any funding-related text from the manuscript and let us know how you would like to update your Funding Statement. Currently, your Funding Statement reads as follows: "Funding was provided to by the Johns Hopkins Hospital Physician Scientist Experiment Fund microgrant (DBL), NINDS R25 (DBL), NIH R01HL139543 (RCK), R01NS099461 (JS), R01NS125653 (JS and SS), R01NS110808 (SS), and R01NS111230 (SS)".

5. Thank you for stating the following in your Competing Interests section:  "NO authors have competing interests".

Please complete your Competing Interests on the online submission form to state any Competing Interests. If you have no competing interests, please state "The authors have declared that no competing interests exist", as detailed online in our guide for authors at http://journals.plos.org/plosone/s/submit-now 

8. We notice that your supplementary table "Primer Table" are included in the manuscript file. Please remove them and upload them with the file type 'Supporting Information'. Please ensure that each Supporting Information file has a legend listed in the manuscript after the references list.

Additional Editor Comments:

Dear author,

Revise the whole manuscript according to both reviewer's comments.

Reviewers' comments:

Reviewer's Responses to Questions

**Comments to the Author**

1. Is the manuscript technically sound, and do the data support the conclusions?

Reviewer #1: Partly

Reviewer #2: Partly

2. Has the statistical analysis been performed appropriately and rigorously? 

Reviewer #1: I Don't Know

Reviewer #2: No

3. Have the authors made all data underlying the findings in their manuscript fully available?

Reviewer #1: Yes

Reviewer #2: Yes

4. Is the manuscript presented in an intelligible fashion and written in standard English?

Reviewer #1: Yes

Reviewer #2: Yes

5. Review Comments to the Author

Reviewer #1: This manuscript describes hippocampal proteome changes in the acute/subacute period after mild hypoxia-asphyxia in the established newborn pig model of neonatal hypoxic-ischemic (H/I) encephalopathy. The study is potentially identifying important pathophysiological mechanisms provided the concerns below are satisfactorily addressed.

Major concerns

1. Based on the experimental design, the sham animals are time-matched for the H/I – 24h but not the H/I 72h animals. One cannot escape the possibility that the sham procedure could cause transient changes in proteomics that remain hidden this way, for instance the changes that are attributed to H/I stress while they may be simply due to the different time-point. A time matched sham group would be welcome.

2. Only female piglets were used. Any particular reason for that? Is the H/I stress employed in the present study has a sex-dependent outcome severity?

3. Throughout the manuscript, the word significant is overused. Understandably, the selection of p<0.1 increases the chance to find statistically significant changes despite the small group n-s, but even when the statistical significance is below the desired 0.05 – calling the changes significant is ambiguous at best when the effect size is minimal. For instance, Table 1 (Significantly upregulated proteins at both timepoints), abundance ratio of the protein PLD4 is 1.106 at 24h, p=0.029. Do the authors honestly believe that this is “significant”. What are the confidence intervals here – how close are those to 1.0 must be. And this is not the only example, only in Table 1 7 of the 9 entries reflect quite small changes. The manuscript should be in my opinion critically revised to showcase those findings where true biological significance is suspected, especially where the effect size is small, and the p>0.05!

Minor issues:

line 72 perhaps “domestic pig” is preferable to “piglet”

line 93 SomnaSol dosage and manufacturer should be added

Figure 1 – glucose levels would be preferred to be given in mmol/L – would make the comparison to lactate ore straightforward. Also “cBase” is unexplained in the Figure legend, likely refers to Base Deficit (negative Base Excess). Clarification would be welcome

line 121 it is somewhat elusive how precisely 11 samples were created for isobaric mass tag labeling – group n-s are undisclosed at this point

line 159 font size gets reduced – why the fine print?

line 173-175 dilution of the antibodies used for western blotting should be disclosed

line 196 - n-s could be disclosed at the beginning of the description of the experiments (and/or also shown in Figure 1), it would make understanding of methods easier (sucha as at line 121 )

line 215 base deficit likely -10.0 +/- 0.7 ?

line 220 the number of proteins identified is: 1,0087. Is it 10,087 or 1,087? Please clarify.

line 255 “associated with glia, neurons and interneurons” Please explain, in broader terms all hippocampal neurons are “interneurons”, what is meant precisely with this distinction

Reviewer #2: This piglet study of mild hypoxic ischemic encephalopathy uses an unbiased proteomics approach and finds changes relevant to inflammation and energy metabolism.

The rationale for pursuing this investigation is that half of HIE cases are mild and over a third of mild cases display long term effects including cognitive delay at 5 years old. It is important to note that despite the hippocampus does not appear overtly injured it is nonetheless smaller in mild injury. In addition, therapies for mild HIE are lacking and mechanisms are understudied.

Here they use a piglet model of HI which is a strength because piglet brains are more similar to humans. Hippocampal tissue was collected at 24h or 72h after injury, protein expression assessed by quantitative liquid chromatography tandem mass spectrometry and pathway analysis was performed. Acylcarnitines were measured separately.

The piglet model is a strength of this project as their brains are more similar to human. The injury was performed at P3 and involved hypoxia (13% O2 x 45 min to yield sats of 50%) followed anoxia (endotracheal tube clamping x 6 min), then recovery (increased O2 to 50% x 5 min, then 25% O2 until sats are 100%), then extubated. Ischemia is not part of the model. N=3 sham controls, 4 24h injured, 4 72h injured. The model may be a little more severe than mild with mean lactate 7.5, mean pH 7.1 and mean base deficit -10.

They chose a p-value curoff = 0.10. By their criteria, 273 proteins changed/increased, 185 at 24h, 88 at 72h, 19 at both 24h and 72h. Despite the small sample size there was clear clustering of the groups by PCA. The heat map showed overall clear changes at 24h and normalization at 72h, although some proteins were only altered at 24h or only at 72h and a few changed at 24h that persisted through 72h. The authors conclude that amino acid, carbohydrate and one-carbon metabolism increased but fat metabolism and oxidative phosphorylation decreased at 24h. Oxphos decreased further at 72h. The authors also present data that canonical pathways for oxphos are not disrupted showing changes in mRNA transcript and protein do not coincide for PDHA1 and LDHA. They also present data that blood acylcarnitine changes at 24h but normalizes by 72h while hippocampal acylcarnitine does not change, trend toward increased short and medium chain by 72h.

Major revisions:

Decrease p-value to 0.05. This would still allow conclusions that changes represented increased inflammation with involvement of endothelial cells and microglia in an immune response. It would also still allow the conclusion that decreased proteins are associated with interneurons, neurons, astrocytes and oligodendrocytes.

For pathways common to the 24h or the 72h timepoint the similar conclusions might still hold true. This may also be true for metabolic pathways at 24h, fatty acid metabolism and amino acid metabolism at 24h, and mitochondrial pathways at 72h.

Minor revisions:

Please discuss how P3 piglet is an appropriate model for newborn human.

Please discuss age at which sham animals were sacrificed.

Please discuss how P3 piglet equivalent to newborn human brain.

Please discuss how this model does not include ischemia.

Please indicate whether resuscitation involved other actions than unclamping ETT and increasing FiO2.

Please include citations for protein functions mentioned throughout the Results section.

Line 324-325 – change to non-canonical proteins involved in mitochondrial processes only (not inflammation)

Line 422 – specify in blood

6. PLOS authors have the option to publish the peer review history of their article (what does this mean? ). If published, this will include your full peer review and any attached files.

**Do you want your identity to be public for this peer review?** For information about this choice, including consent withdrawal, please see our Privacy Policy .

Reviewer #1: No

Reviewer #2: No

---

## [Author Response · Author response to Decision Letter 1]

21 Jan 2025

Response to Reviewers

We thank both reviewers for their thorough, thoughtful, and constructive comments and critiques. The reviewers were supportive and noted that “The study is potentially identifying important pathophysiological mechanisms” using “a piglet model of HI which is a strength.” In this revised manuscript we provide new experimental results and analyses and address all of the reviewers’ concerns. Please find below a detailed point-by-point response.

Reviewers’ comments:

Reviewer #1: This manuscript describes hippocampal proteome changes in the acute/subacute period after mild hypoxia-asphyxia in the established newborn pig model of neonatal hypoxic-ischemic (H/I) encephalopathy. The study is potentially identifying important pathophysiological mechanisms provided the concerns below are satisfactorily addressed.

We appreciate that the reviewer recognizes the importance of this study in identifying potential pathophysiologic mechanisms following mild HIE. Below, we address each of the concerns raised.

Reviewer #1 Major concerns

1. Based on the experimental design, the sham animals are time-matched for the H/I – 24h but not the H/I 72h animals. One cannot escape the possibility that the sham procedure could cause transient changes in proteomics that remain hidden this way, for instance the changes that are attributed to H/I stress while they may be simply due to the different time-point. A time matched sham group would be welcome.

Domestic pig (Sus scrofa) development is very close to human development, including the brain. Developmental changes in piglet brain evolve over weeks and months, which is in contrast to mouse brain development that occurs on the order of days (1). We acknowledge that our sham group is matched to the 24 hour group; however, developmentally, post-natal day 3 controls will not differ substantially from postnatal day 6 piglets. In fact, brain growth in the piglet slows prenatally and picks back up after about postnatal day 10 (1-3). Respecting the humane animal research principal of the “three R’s,” specifically reduction, sacrificing additional animals towards the sham group to obtain incremental additional data was deemed insufficient justification for such an experiment.

2. Only female piglets were used. Any particular reason for that? Is the H/I stress employed in the present study has a sex-dependent outcome severity?

The aim of our study was to determine whether mild HIE, which is commonly observed in clinical practice, affects the hippocampus – the area of the brain which develops later but is responsible for many cognitive functions including working memory and recall. Numerous pre-clinical and large clinical studies report that males are more likely experience mild HIE rather than females (4-6). Furthermore, males have higher rates of morbidity and mortality compared to females following HIE. Hence, we chose to assess the least affected group – females with mild HIE. Our results show that female piglets with mild HIE – the group presumed to have the most favorable outcome – do have changes in the hippocampus. Future studies using large gyrencephalic animals should be conducted to delineate mechanisms responsible for cognitive deficits in children with mild HIE in a sex-specific manner.

3. Throughout the manuscript, the word significant is overused. Understandably, the selection of p<0.1 increases the chance to find statistically significant changes despite the small group n-s, but even when the statistical significance is below the desired 0.05 – calling the changes significant is ambiguous at best when the effect size is minimal. For instance, Table 1 (Significantly upregulated proteins at both timepoints), abundance ratio of the protein PLD4 is 1.106 at 24h, p=0.029. Do the authors honestly believe that this is “significant”. What are the confidence intervals here – how close are those to 1.0 must be. And this is not the only example, only in Table 1 7 of the 9 entries reflect quite small changes. The manuscript should be in my opinion critically revised to showcase those findings where true biological significance is suspected, especially where the effect size is small, and the p>0.05!

In this study we adhered to large data statistical approaches accepted in the field of untargeted proteomics to identify hippocampal pathways affected following mild HIE. In proteomics, where thousands of proteins are analyzed simultaneously, one cannot apply a simple p-value threshold due to multiple testing. Hence, a False Discovery Rate (FDR) correction method is employed to address the large data set obtained using LC-MS/MS. In this study we employed a 5% FDR confidence threshold, which is a standard threshold for identifying significantly differentially expressed proteins to ensure reliable and robust findings (7-9). In the context of our brain proteomics analysis, which identified thousands of proteins, 5% FDR (0.05) means that there is maximum of 5% chance of a false positive (i.e. incorrectly called significant) across the entire dataset.

While FDR provides an overall picture of differentially expressed proteins in the dataset, subsequent analysis using ANOVA and reporting p-values indicates the significance of the individual proteins within the entire dataset that contains thousands of proteins. Although we used a p-value of <0.1 for pathway analysis, all of the tables include proteins with p-values <0.05 and report the exact p-value for each protein. To the reviewer’s point, to illustrate the difference in the number of proteins available for pathway analysis at a threshold of p<0.05 compared to p<0.1, we have adapted Figure 2 to include a Venn diagram of the number of proteins with p<0.05. We acknowledge that p<0.1 is less stringent than p<0.05. The threshold of p<0.1 is in keeping with large data-set accepted practices. In the initial submission, we validated these findings with RT-PCR and western blot. In this resubmission, we further validate the finding that amino acid metabolism pathways are significant altered by measuring amino acids in the brain tissue (hippocampus). We do not rely solely on proteomics data to draw conclusions.

Here we illustrate why a differentially expressed protein with a p-value between 0.05 and 0.1 is important to consider in pathway analysis: Our proteomics analysis demonstrated that metabolic pathways, specifically metabolism of amino acids, were affected at 24 hours (Fig 3). Based on the 5% FDR threshold, the protein ALDH6A1 is considered significant. This protein has an abundance ratio of 1.068 with a p-value 0.074 at 24 hours post mild HIE (Table 3). ALDH6A1 is an enzyme involved in valine and pyrimidine metabolic pathways (10). We validated the significance of this finding by performing amino acid analysis (Fig 4). Tissue amino acid analysis showed that levels of valine were significantly altered at 24 hours (Sham 0.46 ± 0.32 nmol/g vs. 24 hour HIE 2.33 ± 0.14 nmol/g; p<0.0376).

Reviewer #1 Minor issues:

line 72 perhaps „domestic pig” is preferable to „piglet”

Thank you, this has been now corrected.

line 93 SomnaSol dosage and manufacturer should be added

Thank you, this information now has been added.

Figure 1 – glucose levels would be preferred to be given in mmol/L – would make the comparison to lactate ore straightforward.

Thank you for this suggestion. Although in clinical practice in the United States, blood glucose levels are reported in mg/dL, we have changed the figure and text to mmol/L, which is the corresponding SI unit, to improve accessibility for a larger readership.

Also „cBase” is unexplained in the Figure legend, likely refers to Base Deficit (negative Base Excess). Clarification would be welcome.

Thank you. This has been corrected in the figure and legend.

line 121 it is somewhat elusive how precisely 11 samples were created for isobaric mass tag labeling – group n-s are undisclosed at this point

The group-specific n-s are now reported in section Piglet Model of Mild HIE instead of within the statistical analyses section.

line 159 font size gets reduced – why the fine print?

We apologize for this. Formatting has been corrected.

line 173-175 dilution of the antibodies used for western blotting should be disclosed

These are now reported.

line 196 - n-s could be disclosed at the beginning of the description of the experiments (and/or also shown in Figure 1), it would make understanding of methods easier (sucha as at line 121 )

line 215 base deficit likely -10.0 +/- 0.7 ?

As noted above, n’s are now described in the Piglet Model of Mild HIE section. They have also been added to the legend of Fig 1.

line 220 the number of proteins identified is: 1,0087. Is it 10,087 or 1,087? Please clarify.

This typo has been corrected. There were a total 10,087 proteins identified.

line 255 „associated with glia, neurons and interneurons” Please explain, in broader terms all hippocampal neurons are „interneurons”, what is meant precisely with this distinction

The hippocampus, even at early developmental stages, contains both primary pyramidal neurons and interneurons (11, 12). Additionally, glia, neuron, and interneuron are categories specified in Velmeshev et al., which are described in the text and accompanying table (13).

Reviewer #2: This piglet study of mild hypoxic ischemic encephalopathy uses an unbiased proteomics approach and finds changes relevant to inflammation and energy metabolism.

The rationale for pursuing this investigation is that half of HIE cases are mild and over a third of mild cases display long term effects including cognitive delay at 5 years old. It is important to note that despite the hippocampus does not appear overtly injured it is nonetheless smaller in mild injury. In addition, therapies for mild HIE are lacking and mechanisms are understudied.

Here they use a piglet model of HI which is a strength because piglet brains are more similar to humans. Hippocampal tissue was collected at 24h or 72h after injury, protein expression assessed by quantitative liquid chromatography tandem mass spectrometry and pathway analysis was performed. Acylcarnitines were measured separately.

The piglet model is a strength of this project as their brains are more similar to human. The injury was performed at P3 and involved hypoxia (13% O2 x 45 min to yield sats of 50%) followed anoxia (endotracheal tube clamping x 6 min), then recovery (increased O2 to 50% x 5 min, then 25% O2 until sats are 100%), then extubated. Ischemia is not part of the model. N=3 sham controls, 4 24h injured, 4 72h injured. The model may be a little more severe than mild with mean lactate 7.5, mean pH 7.1 and mean base deficit -10.

They chose a p-value curoff = 0.10. By their criteria, 273 proteins changed/increased, 185 at 24h, 88 at 72h, 19 at both 24h and 72h. Despite the small sample size there was clear clustering of the groups by PCA. The heat map showed overall clear changes at 24h and normalization at 72h, although some proteins were only altered at 24h or only at 72h and a few changed at 24h that persisted through 72h. The authors conclude that amino acid, carbohydrate and one-carbon metabolism increased but fat metabolism and oxidative phosphorylation decreased at 24h. Oxphos decreased further at 72h. The authors also present data that canonical pathways for oxphos are not disrupted showing changes in mRNA transcript and protein do not coincide for PDHA1 and LDHA. They also present data that blood acylcarnitine changes at 24h but normalizes by 72h while hippocampal acylcarnitine does not change, trend toward increased short and medium chain by 72h.

We thank this reviewer for the thorough read and meticulous overview of the manuscript.

Reviewer #2 Major concerns:

Decrease p-value to 0.05. This would still allow conclusions that changes represented increased inflammation with involvement of endothelial cells and microglia in an immune response. It would also still allow the conclusion that decreased proteins are associated with interneurons, neurons, astrocytes and oligodendrocytes.

For pathways common to the 24h or the 72h timepoint the similar conclusions might still hold true. This may also be true for metabolic pathways at 24h, fatty acid metabolism and amino acid metabolism at 24h, and mitochondrial pathways at 72h.

As noted in the response to reviewer 1 above, Figure 2 has been adapted to show the number of proteins reaching the threshold of p<0.05. All tables include proteins with a p-value <0.05. In the above response, we outline the rationale for the accepted practice of utilizing a p-value of <0.1 for large proteomics data set pathway analysis.

Reviewer #2 Minor concerns:

Please discuss how P3 piglet is an appropriate model for newborn human.

We have a long-standing history of utilizing an established moderate-severe HIE piglet model. Piglet HIE models use a range of ages (14-18). P3 is the youngest age available, as piglets are not born in our facility and need to acclimate prior to use.

Please discuss age at which sham animals were sacrificed.

Sham animals were sacrificed at 24 hours. This is reported in Methods section.

Please discuss how P3 piglet equivalent to newborn human brain.

Please see the response to Reviewer #1 above. In brief, domestic pig (Sus scrofa) development is very close to human development, including the brain. Developmental changes in piglet brain evolve over weeks and months, which is in contrast to mouse brain development that occurs on the order of days (1).

Please discuss how this model does not include ischemia.

Ischemia in models of HIE refers to cessation of blood flow, which is often achieved by occlusion of carotid arteries in large animals or ligation in smaller animals such as rodents.

Please indicate whether resuscitation involved other actions than unclamping ETT and increasing FiO2.

No other actions, such as administration of epinephrine, bicarbonate, calcium gluconate, or other interventions to augment blood flow and cardiac output or to buffer metabolic acidosis were performed.

Please include citations for protein functions mentioned throughout the Results section.

Citations are included in the text.

Line 324-325 – change to non-canonical proteins involved in mitochondrial processes only (not inflammation)

This is now changed to the following: “….that the proteomics data demonstrates disruption of non-canonical proteins involved in mitochondrial processes and changes in proteins involved in inflammation.”

Line 422 – specify in blood

Thank you. We now made this change.

References:

1. Dobbing J, Sands J. Comparative aspects of the brain growth spurt. Early Hum Dev. 1979;3(1):79-83.

2. Mudd AT, Dilger RN. Early-Life Nutrition and Neurodevelopment: Use of the Piglet as a Translational Model. Adv Nutr. 2017;8(1):92-104.

3. Dickerson JW, Dobbing J. Prenatal and postnatal growth and development of the central nervous system of the pig. Proc R Soc Lond B Biol Sci. 1967;166(1005):384-95.

4. Mirza MA, Ritzel R, Xu Y, McCullough LD, Liu F. Sexually dimorphic outcomes and inflammatory responses in hypoxic-ischemic encephalopathy. J Neuroinflammation. 2015;12:32.

5. Chalak LF, Pruszynski JE, Spong CY. Sex Vulnerabilities to Hypoxia-Ischemia at Birth. JAMA Netw Open. 2023;6(8):e2326542.

6. Kelly LA, Branagan A, Semova G, Molloy EJ. Sex differences in neonatal brain injury and inflammation. Front Immunol. 2023;14:1243364.

7. Ting L, Cowley MJ, Hoon SL, Guilhaus M, Raftery MJ, Cavicchioli R. Normalization and statistical analysis of quantitative proteomics data generated by metabolic labeling. Mol Cell Proteomics. 2009;8(10):2227-42.

8. Boeddrich A, Haenig C, Neuendorf N, Blanc E, Ivanov A, Kirchner M, et al. A proteomics analysis of 5xFAD mouse brain regions reveals the lysosome-associated protein Arl8b as a candidate biomarker for Alzheimer's disease. Genome Med. 2023;15(1):50.

9. Wingo AP, Liu Y, Gerasimov ES, Gockley J, Logsdon BA, Duong DM, et al. Integrating human brain proteomes with genome-wide association data implicates new proteins in Alzheimer's disease pathogenesis. Nat Genet. 2021;53(2):143-6.

10. Marcadier JL, Smith AM, Pohl D, Schwartzentruber J, Al-Dirbashi OY, Majewski J, et al. Mutation

---

## [Decision Letter · Decision Letter 1]

26 Feb 2025

Proteomic analysis of hippocampus reveals metabolic reprogramming in a piglet model of mild hypoxic ischemic encephalopathy

PONE-D-24-22507R1

Dear Dr. Joseph Scafidi.,

We’re pleased to inform you that your manuscript has been judged scientifically suitable for publication and will be formally accepted for publication once it meets all outstanding technical requirements.

Kind regards,

Samiullah Khan, Ph. D

Academic Editor

PLOS ONE

Additional Editor Comments (optional):

Reviewers' comments:

Reviewer's Responses to Questions

**Comments to the Author**

1. If the authors have adequately addressed your comments raised in a previous round of review and you feel that this manuscript is now acceptable for publication, you may indicate that here to bypass the “Comments to the Author” section, enter your conflict of interest statement in the “Confidential to Editor” section, and submit your "Accept" recommendation.

Reviewer #1: All comments have been addressed

2. Is the manuscript technically sound, and do the data support the conclusions?

Reviewer #1: Yes

3. Has the statistical analysis been performed appropriately and rigorously? 

Reviewer #1: Yes

4. Have the authors made all data underlying the findings in their manuscript fully available?

Reviewer #1: Yes

5. Is the manuscript presented in an intelligible fashion and written in standard English?

Reviewer #1: Yes

6. Review Comments to the Author

Reviewer #1: The authors have addressed the concerns raised in the review, justified their approach and/or modified the manuscript to comply with suggestions. No further issues remain, I'd take the opportunity to congratulate the authors for their achievements with this manuscript

7. PLOS authors have the option to publish the peer review history of their article (what does this mean? ). If published, this will include your full peer review and any attached files.

**Do you want your identity to be public for this peer review?** For information about this choice, including consent withdrawal, please see our Privacy Policy .

Reviewer #1: No

---

## [Editor Report · Acceptance letter]

PONE-D-24-22507R1

PLOS ONE

Dear Dr. Scafidi,

I'm pleased to inform you that your manuscript has been deemed suitable for publication in PLOS ONE. Congratulations! Your manuscript is now being handed over to our production team.

Kind regards,

on behalf of

Dr. Samiullah Khan

Academic Editor

PLOS ONE